# Ranking Recovery under Privacy Considerations

**Minoh Jeong**                                                                  *jeong316@umn.edu*
*Department of Electrical and Computer Engineering*
*University of Minnesota*

**Alex Dytso**                                                                  *alex.dytso@njit.edu*
*Department of Electrical and Computer Engineering*
*New Jersey Institute of Technology*

**Martina Cardone**                                                              *mcardone@umn.edu*
*Department of Electrical and Computer Engineering*
*University of Minnesota*

**Reviewed on OpenReview:** *https://openreview.net/forum?id=2EOVIvRXlv*

## Abstract

We consider the *private ranking recovery problem*, where a data collector seeks to estimate the permutation/ranking of a data vector given a randomized (privatized) version of it. We aim to establish fundamental trade-offs between the performance of the estimation task, measured in terms of *probability of error*, and the *level of privacy* that can be guaranteed when the noise mechanism consists of adding artificial noise. Towards this end, we show the optimality of a low-complexity decision rule (referred to as linear decoder) for the estimation task, under several noise distributions widely used in the privacy literature (e.g., Gaussian, Laplace, and generalized normal model). We derive the Taylor series of the probability of error, which yields its first and second-order approximations when such a linear decoder is employed. We quantify the guaranteed level of privacy using differential privacy (DP) types of metrics, such as $\epsilon$-DP and $(\alpha, \epsilon)$-Rényi DP. Finally, we put together the results to characterize trade-offs between privacy and probability of error.

## 1 Introduction

Today, *ranking* algorithms are of fundamental importance and are used in a large portfolio of applications, such as search engines (Dwork et al., 2001), biomedical (Chanas & Kobylański, 1996), recommender systems (Baskin & Krishnamurthi, 2009), and feature matching (Jeong et al., 2020). Broadly speaking, the goal of a ranking algorithm is to sort a dataset – which, in the current Big Data era, is usually massive in size and complexity – so that users/individuals are provided with accurate and relevant results. For instance, a recommender system may suggest a new item to buy to a user based on their interests and previous purchases. Although modern ranking algorithms promise efficient means of performing large-scale data processing, there are numerous *privacy* considerations that must not be overlooked. For instance, the dataset might contain confidential data, such as clinical/genomic health and banking records, or a user would not like to disclose their previous purchases to a recommender system.

In this paper, we study the *private ranking recovery* problem, which consists of recovering the ranking/permutation of an input data vector from a noisy version of it. The importance and timeliness of this problem stems from two major considerations. First, many modern computing systems (e.g., recommender systems) are often more interested in recovering the permutation, i.e., the relative ranking of data points, rather than the values of the data itself. Second, because of privacy considerations, users might decide to privatize their data (e.g., by adding some noise) before sharing it with an external party (e.g., recommender system). These facts give rise to the following practically relevant question: *Which perturbation*

*mechanisms allow for data privatization, while still allowing to correctly recover the permutation of the input data vector with high probability?*

## 1.1 Related Work

Recently, problems with a similar flavor to the private ranking recovery problem (studied in this paper) have been analyzed. For instance, Jeong et al. (2020; 2021) focused on characterizing the probability of error of estimating the original permutation of data perturbed by adding Gaussian noise. In particular, they characterized sufficient and necessary conditions of the Gaussian noise covariance matrix that ensure that the optimal (i.e., that minimizes the probability of error) decision rule consists of simply declaring a permutation-independent linear function of the noisy observation. The rank aggregation problem was studied under differential privacy constraints by Shang et al. (2014) and Hay et al. (2017), and under local differential privacy by Yan et al. (2020) and Alabi et al. (2021). *Differential privacy* (DP), which is a statistical guarantee introduced by Dwork et al. (2006b) for indistinguishability whether any data element exists or not in a dataset, is one of the most common adopted privacy metrics. Several notions of DP have been introduced and analyzed that range from the basic $\epsilon$-DP metric (which can be guaranteed by using the Laplace randomized mechanism) (Dwork, 2008), to more relaxed versions of it, such as the $(\epsilon, \delta)$-DP Dwork et al. (2006a), the $\epsilon$-mutual information DP and the $\epsilon$-Kullback-Leibler (KL) DP Cuff & Yu (2016), and the $(\alpha, \epsilon)$-Rényi DP (RDP) Mironov (2017). In particular, the $(\alpha, \epsilon)$-RDP encompasses: (i) the $\epsilon$-DP if $\alpha \to \infty$, and (ii) the $\epsilon$-KL DP if $\alpha \to 1$. Moreover, some important properties of the $\epsilon$-DP, e.g., the composition theorem, remain applicable in the RDP framework.

When data is confidential and hence, needs to be privatized before being shared with an external party (which will perform some operations on it) a natural question arises: *For a fixed target performance guarantee (a.k.a. utility) required on the data, what is a randomized mechanism that achieves the maximum level of privacy?* To answer this question, one needs to understand the *trade-off* between privacy and utility. Such a trade-off has been studied in the literature in several settings, where different utility measures have been used. For instance, Wasserman & Zhou (2010) compared several randomized mechanisms (from a statistical point of view) by using the Kolmogorov–Smirnov and the $L_2$ distances among distributions and densities. Wei et al. (2020) showed a trade-off between the convergence of a federated learning algorithm (utility) and the level of privacy (measured in terms of DP) that can be guaranteed, hence suggesting the amount of artificial noise that should be used in this context. Avent et al. (2019) studied the privacy-utility trade-off for a hyperparameterized algorithm using multi-objective optimization and Pareto front. For a single real-valued query, Geng et al. (2015) identified the staircase distribution (i.e., a geometric mixture of uniform random variables) as a distribution that minimizes the $L_1$ loss (utility) under a fixed given level of $\epsilon$-DP. The staircase distribution has also been shown to be the optimal $\epsilon$-DP mechanism under other utility constraints (Soria-Comas & Domingo-Ferrer, 2013). More recently, Geng et al. (2019) studied trade-offs between $(0, \delta)$-DP and the $L_p$ loss function for a single real-valued query function.

## 1.2 Contributions and Paper Organization

In this paper, we study the *private ranking recovery problem*, where a confidential input data vector needs to be privatized (by means of a randomized mechanism) before being shared with an external party. We seek to retrieve the true ordering of the input data vector given the privatized data. Our main goal is to characterize the trade-off between the performance of estimating the permutation of the input data vector (measured in terms of *error probability*) and the level of *privacy* (measured in terms of $\epsilon$-DP and $(\alpha, \epsilon)$-RDP) that the used mechanism guarantees.

First, in Section 2, we formulate the private ranking recovery problem within a DP framework. In particular, we adopt the $(\alpha, \epsilon)$-RDP as a privacy metric; our choice mainly stems from the fact that the $(\alpha, \epsilon)$-RDP encompasses other widely employed DP metrics such as the $\epsilon$-DP (Dwork, 2008) if $\alpha \to \infty$, and the $\epsilon$-KL DP (Cuff & Yu, 2016) if $\alpha \to 1$. Moreover, as pointed out in (Mironov, 2017, Proposition 3), $(\alpha, \epsilon)$-RDP can be converted to $(\epsilon, \delta)$-DP.

Second, in Section 3, we show that under mild assumptions on the input data vector (i.e., the input data distribution is exchangeable) and on the randomized mechanism (i.e., it has an $\ell_p$-spherical distribution),

Table 1: Trade-off between privacy and utility in the low-noise regime with i.i.d. noise components. Privacy is measured by $(\alpha, \epsilon)$-RDP for the Gaussian and Laplace mechanisms and by $\epsilon$-DP for the generalized normal mechanism. The utility is quantified by $P_e$.

| $\mathcal{K}(\sigma)$ | **Trade-off** |
|---|---|
| $\mathcal{N}(0,1)$ | $P_e \propto \left(\frac{\alpha}{\epsilon}\right)^{1/2}$ |
| $\mathrm{Lap}\left(0, \frac{1}{\sqrt{2}}\right)$ | $P_e \propto \frac{1}{\epsilon}$ |
| $\mathcal{GN}\left(0, \sqrt{\frac{\Gamma(p^{-1})}{\Gamma(3p^{-1})}}, p\right)$ | $P_e \propto \left(\frac{1}{\epsilon}\right)^{1/p}$ |

declaring the permutation of the observed noisy vector is an optimal decision rule for recovering the permutation of the input data vector. Because of this, and using the terminology introduced in Jeong et al. (2020; 2021), we refer to such a decision rule as *linear decoder*. This has complexity $O(n \log n)$, which is a significant reduction with respect to the $O(n!)$ complexity of a naive brute-force implementation of the optimal decoder.

Third, in Section 3, we characterize the error probability of the linear decoder, by deriving the Taylor series of it. This result suggests that the private ranking recovery problem is noise dominated, i.e., the error probability is large even for small values of the noise variance. Further, we derive the first-order approximation of the error probability with respect to the noise standard deviation, and we verify through numerical simulations that this approximation is indeed accurate. In particular, our first-order approximation expression decouples the effects of the input data distribution and noise distribution on the error probability. We also derive the exact expression for the linear slope of the error probability for the case of i.i.d. input data vector entries.

Finally, in Section 4 we derive the trade-off between privacy (measured by $\epsilon$-DP and $(\alpha, \epsilon)$-RDP) and utility (measured by the error probability $P_e$) in the low-noise regime. We consider widely used noise addition mechanisms, i.e., the Laplace, the Gaussian, and the generalized normal. As indicated in Table 1, these mechanisms have different relationships[1] between $\epsilon$ and $P_e$. The trade-offs for $\mathcal{N}(0,1)$ and $\mathrm{Lap}\left(0, \frac{1}{\sqrt{2}}\right)$ are obtained based on $(\alpha, \epsilon)$-RDP, and for the generalized normal mechanism with $p \leq 1$, $\epsilon$-DP is considered. We observe that the generalized normal mechanism with $p \leq 1$ offers the best trade-off.

### 1.3 Notation

Boldface upper case letters $\mathbf{X}$ denote random vectors; the boldface lower case letter $\mathbf{x}$ indicates a specific realization of $\mathbf{X}$; $X_{i:n}$ denotes the $i$-th order statistics of $\mathbf{X} \in \mathbb{R}^n$; $[n_1 : n_2]$ is the set of integers from $n_1$ to $n_2 \geq n_1$; $I_n$ is the identity matrix of dimension $n$; $\mathbf{0}_n$ is the column vector of dimension $n$ of all zeros; $\|\mathbf{x}\|_p$ denotes the $\ell_p$-norm of $\mathbf{x} \in \mathbb{R}^n$; calligraphic letters indicate sets; $|\mathcal{A}|$ is the cardinality of $\mathcal{A}$. $\mathbf{X}_{\mathcal{A}}$ is the subvector of $\mathbf{X}$ where only the elements in $\mathcal{A}$ are retained. For two $n$-dimensional vectors $\mathbf{x}$ and $\mathbf{y}$, if for all $i \in [1 : n]$, the $i$-th element of $\mathbf{x}$ is larger than or equal to the $i$-th element of $\mathbf{y}$, then we use $\mathbf{x} \geq \mathbf{y}$; $\mathbb{1}_{\mathcal{S}}$ is the indicator function over the set $\mathcal{S}$. For any $\mathbf{x} \in \mathbb{R}^n$ and $\mathbf{y} \in \mathbb{R}^n$, the Hamming distance is defined as $d_H(\mathbf{x}, \mathbf{y}) = \sum_{i=1}^n \mathbb{1}_{\{x_i \neq y_i\}}$; $\overset{d}{=}$ denotes equality in distribution; $\mathcal{N}(\boldsymbol{\mu}_n, K)$ is the $n$-dimensional Gaussian distribution with mean $\boldsymbol{\mu}_n$ and covariance matrix $K$; $\mathrm{Lap}(\mu, b)$ is the Laplace distribution with mean $\mu$ and scale $b$; $\mathcal{GN}(\mu, a, p)$ is the generalized normal distribution (Nadarajah, 2005; Dytso et al., 2018) with mean $\mu$, scale $a$, and shape $p$. We let $\mathcal{P}$ be the set of all permutations of an $n$-dimensional vector. For $\tau \in \mathcal{P}$, we define

$$\mathcal{H}_\tau = \{\mathbf{x} \in \mathbb{R}^n : x_{\tau_1} \leq x_{\tau_2} \leq \cdots \leq x_{\tau_n}\}, \tag{1}$$

---

[1]The probability of error $P_e$ is proportional up to the first-order term $\left(\frac{1}{\epsilon}\right)^{1/p}$.

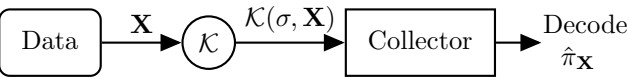

Figure 1: Graphical representation of the considered private ranking recovery framework.

with $x_{\tau_i}, i \in [1:n]$ being the $\tau_i$-th element of $\mathbf{x}$, and $\tau_i, i \in [1:n]$ being the $i$-th element of $\tau$. For example, in the 3-dimensional space there exist $|\mathcal{P}| = 6$ permutations, and we have

$$\begin{aligned} \mathcal{H}_{(1,2,3)} &: X_1 \leq X_2 \leq X_3, \quad \mathcal{H}_{(1,3,2)} : X_1 \leq X_3 \leq X_2, \\ \mathcal{H}_{(2,1,3)} &: X_2 \leq X_1 \leq X_3, \quad \mathcal{H}_{(2,3,1)} : X_2 \leq X_3 \leq X_1, \\ \mathcal{H}_{(3,1,2)} &: X_3 \leq X_1 \leq X_2, \quad \mathcal{H}_{(3,2,1)} : X_3 \leq X_2 \leq X_1, \end{aligned}$$

where $X_i, i \in [1:3]$ is the $i$-th element of $\mathbf{X}$.

## 2 Problem Formulation

We consider the private ranking recovery problem, as shown in Figure 1. In this setting, because of privacy considerations, a randomized mechanism $\mathcal{K}(\cdot)$ is applied on the confidential $n$-dimensional data vector $\mathbf{X} \in \mathbb{R}^n$, before this data is collected by an external party (e.g., recommender system). In other words, $\mathcal{K}(\cdot)$ is applied so as to hide the values of $\mathbf{X}$ from the collector (i.e., privatize $\mathbf{X}$). The goal of the data collector is then to retrieve the permutation $\pi_{\mathbf{X}}$ according to which $\mathbf{X}$ is sorted, i.e., to output the estimate $\hat{\pi}_{\mathbf{X}}$.

In the framework described above, a natural trade-off arises between the performance of the estimation task, referred to as utility function in the remaining of this paper, and the privacy level that can be guaranteed. In particular, such a trade-off is dictated by the distribution of $\mathbf{X}$, and $\mathcal{K}(\cdot)$. In this work, we are interested in characterizing such a trade-off for randomized mechanisms that consist of noise addition on the data vector $\mathbf{X}$, namely

$$\mathcal{K}(\sigma, \mathbf{X}) \triangleq \mathbf{X} + \sigma \mathbf{N}, \tag{2}$$

where $\mathbf{N} \in \mathbb{R}^n$ is the $n$-dimensional noise random vector and $\sigma \geq 0$ is a parameter controlling the power of the noise.

**Utility Function.** As utility function, we consider the *probability of error* incurred in the estimation of $\pi_{\mathbf{X}}$. With reference to Figure 1, we let $\phi(\cdot) : \mathbb{R}^n \to \mathcal{P}$ denote the decoder that the data collector uses to output $\hat{\pi}_{\mathbf{X}}$. Then, the probability of error of the estimation task depends both on $\phi(\cdot)$ and $\mathcal{K}(\cdot)$, that is

$$P_e(\phi, \mathcal{K}) = \Pr\left(\phi(\mathcal{K}(\sigma, \mathbf{X})) \neq \pi_{\mathbf{X}}\right). \tag{3}$$

**Privacy Metric.** Given $\mathcal{K}(\sigma, \mathbf{X})$ in (2), it is important to quantify the privacy level guaranteed by this mechanism. Towards this end, we leverage the $\epsilon$-DP (Dwork, 2008) in Definition 2.1 and the $(\alpha, \epsilon)$-RDP (Mironov, 2017) in Definition 2.2.

**Definition 2.1.** Let $\mathcal{X}$ be the set of possible $n$-dimensional real-valued data vectors. Let $(\mathbf{X}, \tilde{\mathbf{X}}) \in \mathcal{X}^2$ be a pair of adjacent data vectors, which differ in at most one element, i.e., $d_H(\mathbf{X}, \tilde{\mathbf{X}}) \leq 1$. Then, the randomized mechanism $\mathcal{K}(\cdot)$ gives $\epsilon$-DP if, for any set $\mathcal{S}$, we have that

$$\Pr(\mathcal{K}(\sigma, \mathbf{X}) \in \mathcal{S}) \leq e^\epsilon \Pr(\mathcal{K}(\sigma, \tilde{\mathbf{X}}) \in \mathcal{S}). \tag{4}$$

**Definition 2.2.** Let $\mathcal{X}$ be the set of possible $n$-dimensional real-valued data vectors. Let $(\mathbf{X}, \tilde{\mathbf{X}}) \in \mathcal{X}^2$ be a pair of adjacent data vectors, which differ in at most one element, i.e., $d_H(\mathbf{X}, \tilde{\mathbf{X}}) \leq 1$. Then, for $\alpha \geq 1$, the randomized mechanism $\mathcal{K}(\cdot)$ gives $(\alpha, \epsilon)$-RDP if

$$\mathsf{RDP}_\alpha(\mathcal{K}) \leq \epsilon, \tag{5a}$$

where

$$\mathsf{RDP}_\alpha(\mathcal{K}) = \sup_{(\mathbf{X}, \tilde{\mathbf{X}}) \in \mathcal{X}^2 : d_H(\mathbf{X}, \tilde{\mathbf{X}}) \leq 1} D_\alpha(\mathcal{K}(\sigma, \mathbf{X}) \| \mathcal{K}(\sigma, \tilde{\mathbf{X}})), \tag{5b}$$

and, for $\mathbf{X}$ and $\mathbf{Y}$ with equal support,

$$D_\alpha(\mathbf{X}\|\mathbf{Y}) = \frac{1}{\alpha - 1} \log \mathbb{E}\left[\left(\frac{f_\mathbf{X}(\mathbf{Y})}{f_\mathbf{Y}(\mathbf{Y})}\right)^\alpha\right], \tag{5c}$$

with $f_\mathbf{X}(\cdot)$ and $f_\mathbf{Y}(\cdot)$ being the probability density functions (PDFs) of $\mathbf{X}$ and $\mathbf{Y}$, respectively. $D_\alpha(\cdot\|\cdot)$ is the Rényi divergence of order $\alpha$.

Several rationales are behind our choice of using the $(\alpha, \epsilon)$-RDP as a privacy measure. First, the $(\alpha, \epsilon)$-RDP encompasses other widely employed DP metrics, e.g., the $\epsilon$-DP (Dwork, 2008) if $\alpha \to \infty$, and the $\epsilon$-KL DP (Cuff & Yu, 2016) if $\alpha \to 1$. The $(\alpha, \epsilon)$-RDP also bypasses some limitations of the $\epsilon$-DP (e.g., a Gaussian noise adding mechanism is not $\epsilon$-DP), while still retaining similar appealing properties (e.g., composition properties (Mironov, 2017)) as those of the $\epsilon$-DP.

Our goal in this paper is to characterize the privacy-utility trade-off when the randomized mechanism in (2) is used. In other words, we seek to determine $P_e(\phi, \mathcal{K})$ in (3), subject to the constraint that $\mathsf{RDP}_\alpha(\mathcal{K})$ in (5) is set to be equal to $\epsilon$ (for $\epsilon$-DP we set $\alpha = \infty$). In particular, we will focus on scenarios where $\mathbf{X}$ is exchangeable and $\mathbf{N} \in \mathcal{S}_{n,p}$, as defined below.

**Definition 2.3.** A sequence of random variables $X_1, \ldots, X_n$ is said to be exchangeable if, for any permutation $\pi = (\pi_1, \ldots, \pi_n)$ of $[1:n]$, we have

$$(X_1, \ldots, X_n) \stackrel{d}{=} (X_{\pi_1}, \ldots, X_{\pi_n}).$$

**Definition 2.4.** A function $f$ is $\ell_p$-spherically non-increasing if it can be written as

$$f(\mathbf{x}) = g(\|\mathbf{x}\|_p), \tag{6}$$

where $g : \mathbb{R}_+ \to \mathbb{R}_+$ is a non-increasing function. We denote by $\mathcal{S}_{n,p}$ the set of $n$-dimensional distributions which have an $\ell_p$-spherically non-increasing density function.

Our assumption on $\mathbf{X}$ being exchangeable includes data that does not need to be necessarily i.i.d., but can be correlated. For instance, any convex combination of i.i.d. random variables, and any spherically contoured distribution are exchangeable[2]. We also highlight that $\mathbf{N} \in \mathcal{S}_{n,p}$ implies that $\mathbf{N}$ is exchangeable; this follows since the $\ell_p$-norm is permutation invariant. Finally, we conclude this section with a few examples (see Appendix A for the details), which show that distributions on $\mathbf{N}$ widely used in the DP literature are in $\mathcal{S}_{n,p}$. Thus, the assumption that $\mathbf{N} \in \mathcal{S}_{n,p}$ can also be considered as mild.

*Example* 2.5. The following distributions belong to $\mathcal{S}_{n,p}$:

- $\mathbf{N} \sim \mathcal{N}(\mathbf{0}_n, \sigma^2 I_n)$: in this case, $p = 2$;
- $\mathbf{N}$ consists of i.i.d. $\mathrm{Lap}(0, b)$: in this case, $p = 1$;
- $\mathbf{N}$ consists of i.i.d. $\mathcal{GN}(0, a, p)$;
- $\mathbf{N}$ has a staircase distribution (Geng et al., 2015): in this case, $p = 1$;
- $\mathbf{N} \sim \mathrm{Unif}\left(\mathcal{B}_p(\mathbf{0}_n, r)\right)$ with $r > 0$, where $\mathcal{B}_p(\mathbf{0}_n, r) = \{\mathbf{x} \in \mathbb{R}^n : \|\mathbf{x}\|_p < r\}$ is the $\ell_p$-ball centered at $\mathbf{0}_n$.

# 3 Accuracy of Ranking Recovery

In this section, we seek to derive an expression for the probability of error of estimating $\pi_\mathbf{X}$. In Section 3.1, we first revisit a low-complexity decoder, and show its optimality under the assumptions of Section 2. Then, in Section 3.2 we characterize $P_e(\phi, \mathcal{K})$ for this decoder. In Section 3.3, we derive an accurate first-order approximation of $P_e(\phi, \mathcal{K})$, which we will leverage to characterize the privacy-utility trade-offs. Finally, in Section 3.4, we evaluate the derived first-order approximation of $P_e(\phi, \mathcal{K})$ for the case when the data $\mathbf{X}$ is i.i.d. and $n$ is large (i.e., the high-dimensional regime).

---

[2]This restriction can be thought of as a limitation of our results. However, to make progress on this problem in a Bayesian framework, making assumptions is eventually inevitable as otherwise, the problem becomes intractable, and one will not be able to say much about the limits of permutation recovery. Assuming an exchangeable data distribution is reasonable whenever the data has no natural order. A particular example is relational data such as social network users, ratings, and preference data (Lloyd et al., 2012). The exchangeability assumption, in our opinion, strikes a good balance between how permutation recovery would behave in practice and the problem theoretical solvability.

### 3.1 Optimal Decoder with Low-Complexity

As illustrated in Section 2, the data collector uses a decoder $\phi(\cdot) : \mathbb{R}^n \to \mathcal{P}$ to output $\hat{\pi}_{\mathbf{X}}$. In what follows, we let $\phi_{\text{opt}}(\cdot)$ denote the *optimal* decoder, i.e., the decoder that recovers $\hat{\pi}_{\mathbf{X}}$ such that the probability of error defined in (3) is minimized. We also consider a (potentially sub-optimal) decoder to which (borrowing the terminology used by Jeong et al. (2020; 2021)) we refer as *linear decoder* and formally define below.

**Definition 3.1.** Given the noisy data vector $\mathbf{y} \in \mathbb{R}^n$, the linear decoder is defined as

$$\phi_{\text{lin}}(\mathbf{y}) = \pi_{\mathbf{y}}, \tag{7}$$

where $\pi_{\mathbf{y}}$ denotes the permutation according to which $\mathbf{y}$ is sorted.

The decoder in (7) is a special case of a more general linear decoder $\pi_{A\mathbf{y}+\mathbf{b}}$, where $A \in \mathbb{R}^{n \times n}$ and $\mathbf{b} \in \mathbb{R}^n$; such a linear decoder can be optimal when the noise has memory (Nomakuchi & Sakata, 1988a;b; Jeong et al., 2020). In (7), we set $A = I_n$ and $\mathbf{b} = \mathbf{0}_n$.

The linear decoder $\phi_{\text{lin}}(\cdot)$ has several advantages, among which its low-complexity: it simply consists of a sorting operation and hence, it has a complexity of $O(n \log n)$. This is a significant reduction with respect to the $O(n!)$ complexity of a naive brute-force implementation of the optimal decoder $\phi_{\text{opt}}(\cdot)$ based on the maximum a posteriori (MAP) decision rule (Kay, 1998). Moreover, as we will show in Theorem 3.3, the linear decoder $\phi_{\text{lin}}(\cdot)$ is indeed optimal (i.e., $\phi_{\text{lin}}(\cdot) = \phi_{\text{opt}}(\cdot)$) under the assumptions stated in Section 2. In particular, to show this result we will leverage the following lemma (proof in Appendix B).

**Lemma 3.2.** *For any two $n$-dimensional vectors $\mathbf{x} \in \mathcal{H}_\eta$ and $\mathbf{y} \in \mathcal{H}_\tau$, and $p \geq 1$, we have that*

$$\tau \in \arg\min_{\omega \in \mathcal{P}} \|\mathbf{y} - P_{\eta \to \omega}\mathbf{x}\|_p, \tag{8}$$

*where $P_{\eta \to \omega}$ is the permutation matrix that permutes $\mathbf{x} \in \mathcal{H}_\eta$ into $P_{\eta \to \omega}\mathbf{x} \in \mathcal{H}_\omega$.*

Lemma 3.2 states that, when $p \geq 1$, the $\ell_p$-norm of the difference between two given vectors is minimized when the two vectors are sorted according to the same permutation. Lemma 3.2 allows us to prove our first main result, which is given by the next theorem.

**Theorem 3.3.** *Let $\mathbf{X} \in \mathbb{R}^n$ be exchangeable, and assume that the randomized mechanism $\mathcal{K}(\sigma, \mathbf{X})$ adopts $\mathbf{N} \in \mathcal{S}_{n,p}$, $p \geq 1$. Then, given any noisy data vector $\mathbf{y} \in \mathbb{R}^n$, we have that*

$$\phi_{\text{opt}}(\mathbf{y}) = \phi_{\text{lin}}(\mathbf{y}). \tag{9}$$

*Proof.* Since $\mathbf{X}$ is exchangeable, all hypotheses are equally-likely (i.e., $\Pr(\mathbf{X} \in \mathcal{H}_\tau) = \frac{1}{n!}$, $\forall \tau \in \mathcal{P}$), and the maximum likelihood decoder is optimal (Kay, 1998). This can be shown as follows,

$$\phi_{\text{opt}}(\mathbf{y}) = \arg\max_{\tau \in \mathcal{P}} \Pr(\mathbf{X} \in \mathcal{H}_\tau \mid \mathcal{K}(\sigma, \mathbf{X}) = \mathbf{y})$$

$$= \arg\max_{\tau \in \mathcal{P}} \frac{\Pr(\mathbf{X} \in \mathcal{H}_\tau)}{f_{\mathcal{K}(\sigma, \mathbf{X})}(\mathbf{y})} f_{\mathcal{K}(\sigma, \mathbf{X})}(\mathbf{y} \mid \mathbf{X} \in \mathcal{H}_\tau)$$

$$= \arg\max_{\tau \in \mathcal{P}} f_{\mathcal{K}(\sigma, \mathbf{X})}(\mathbf{y} \mid \mathbf{X} \in \mathcal{H}_\tau), \tag{10}$$

where $f_{\mathcal{K}(\sigma, \mathbf{X})}$ is the PDF of $\mathcal{K}(\sigma, \mathbf{X})$. We note that the second equality follows by the Bayes' rule, and the last equality follows by the facts that $\Pr(\mathbf{X} \in \mathcal{H}_\tau)$ is a constant for all $\tau \in \mathcal{P}$ and that $f_{\mathcal{K}(\sigma, \mathbf{X})}(\mathbf{y})$ is independent of $\tau$. Therefore, given $\mathbf{y} \in \mathbb{R}^n$ for $\mathcal{K}(\sigma, \mathbf{X})$, an optimal decoder is given by

$$\phi_{\text{opt}}(\mathbf{y}) = \arg\max_{\tau \in \mathcal{P}} f_{\mathcal{K}(\sigma, \mathbf{X})}(\mathbf{y} \mid \mathbf{X} \in \mathcal{H}_\tau). \tag{11}$$

Since $\mathbf{X}$ and $\mathbf{N}$ are independent, the conditional density function in (11) can be written as

$$f_{\mathcal{K}(\sigma, \mathbf{X})}(\mathbf{y} \mid \mathbf{X} \in \mathcal{H}_\tau) = \int f_{\mathbf{X}}(\mathbf{x} \mid \mathcal{H}_\tau) f_{\mathbf{N}}(\mathbf{y} - \mathbf{x}) \, d\mathbf{x}$$

$$= n! \int \mathbb{1}_{\{\mathbf{x} \in \mathcal{H}_\tau\}} f_{\mathbf{X}}(\mathbf{x}) g(\|\mathbf{y} - \mathbf{x}\|_p) \, d\mathbf{x}, \tag{12}$$

where in the last equality we used Definition 2.4 with $g(\cdot)$ being a non-increasing function. Similarly, we have

$$f_{\mathcal{K}(\sigma,\mathbf{X})}(\mathbf{y} \mid \mathbf{X} \in \mathcal{H}_\eta) = n! \int \mathbb{1}_{\{\mathbf{x} \in \mathcal{H}_\eta\}} f_{\mathbf{X}}(\mathbf{x}) g(\|\mathbf{y} - \mathbf{x}\|_p) \, \mathrm{d}\mathbf{x}$$

$$= n! \int \mathbb{1}_{\{\mathbf{u} \in \mathcal{H}_\tau\}} f_{\mathbf{X}}(\mathbf{u}) g(\|\mathbf{y} - P_{\tau \to \eta}\mathbf{u}\|_p) \, \mathrm{d}\mathbf{u}, \tag{13}$$

where (13) follows by substituting $\mathbf{x} = P_{\tau \to \eta}\mathbf{u}$.

Now, by taking the difference between (12) and (13), we obtain

$$\frac{1}{n!} \left( f_{\mathcal{K}(\sigma,\mathbf{X})}(\mathbf{y} \mid \mathbf{X} \in \mathcal{H}_\tau) - f_{\mathcal{K}(\sigma,\mathbf{X})}(\mathbf{y} \mid \mathbf{X} \in \mathcal{H}_\eta) \right) = \int_{\mathbf{x} \in \mathcal{H}_\tau} f_{\mathbf{X}}(\mathbf{x}) \left( g(\|\mathbf{y} - \mathbf{x}\|_p) - g(\|\mathbf{y} - P_{\tau \to \eta}\mathbf{x}\|_p) \right) \mathrm{d}\mathbf{x}. \tag{14}$$

Using Lemma 3.2, we have that if $\mathbf{y} \in \mathcal{H}_\tau$, then the integrand in (14) is always non-negative. Hence, for any $\mathbf{y}$ sorted according to $\pi_{\mathbf{y}}$, we have

$$\phi_{\mathrm{opt}}(\mathbf{y}) = \arg\max_{\tau \in \mathcal{P}} f_{\mathcal{K}(\sigma,\mathbf{X})}(\mathbf{y} \mid \mathbf{X} \in \mathcal{H}_\tau) = \pi_{\mathbf{y}} = \phi_{\mathrm{lin}}(\mathbf{y}),$$

where the last equality follows from Definition 3.1. This concludes the proof of Theorem 3.3. $\qquad\square$

*Remark* 3.4. We highlight that Jeong et al. (2020; 2021) showed a similar result as in Theorem 3.3 for the case of Gaussian noise, under some specific conditions on the noise covariance matrix. Theorem 3.3 extends the result on the optimality of the linear decoder beyond Gaussian noise, i.e., whenever $\mathbf{N} \in \mathcal{S}_{n,p}, \ p \geq 1$. In particular, to show this result we have leveraged a completely new proof which uses a generalized version of the rearrangement inequality needed in the proof of Lemma 3.2 (see Appendix B).

## 3.2 Error Analysis for $\phi_{\mathrm{lin}}(\cdot)$

We here characterize the error probability of the low-complexity and optimal (as proved in Theorem 3.3 under some assumptions) decoder $\phi_{\mathrm{lin}}(\cdot)$ in Definition 3.1. From (3), the error probability when $\phi_{\mathrm{lin}}(\cdot)$ is used is

$$P_e(\phi_{\mathrm{lin}}, \mathcal{K}) = \Pr(\phi_{\mathrm{lin}}(\mathcal{K}(\sigma, \mathbf{X})) \neq \pi_{\mathbf{X}}).$$

Before deriving $P_e(\phi_{\mathrm{lin}}, \mathcal{K})$, we first introduce the matrix $T_\tau \in \mathbb{R}^{(n-1) \times n}$, for all $\tau \in \mathcal{P}$, as follows

$$(T_\tau)_{i,j} = \mathbb{1}_{\{j = \tau_{i+1}\}} - \mathbb{1}_{\{j = \tau_i\}}. \tag{15}$$

For instance, let $n = 4$ and $\tau = (4, 2, 1, 3)$; then,

$$T_{(4,2,1,3)} = \begin{bmatrix} 0 & 1 & 0 & -1 \\ 1 & -1 & 0 & 0 \\ -1 & 0 & 1 & 0 \end{bmatrix}.$$

*Remark* 3.5. For any exchangeable $\mathbf{X} \in \mathbb{R}^n$, we have that (Pyke, 1965)

$$T_\tau \mathbf{X} \mid \mathbf{X} \in \mathcal{H}_\tau \stackrel{d}{=} \mathbf{W}, \ \forall \tau \in \mathcal{P}, \tag{16a}$$

where $\mathbf{W} \in \mathbb{R}^{n-1}$ is known as the spacing vector (David & Nagaraja, 2004) with

$$W_i \stackrel{d}{=} X_{i+1:n} - X_{i:n}, \ i \in [1 : n-1], \tag{16b}$$

where $X_{i:n}$ is the $i$-th order statistics of $\mathbf{X}$.

The theorem below provides an expression for the error probability of the private ranking recovery problem when the linear decoder $\phi_{\mathrm{lin}}(\cdot)$ in Definition 3.1 is used. In particular, this expression is derived by considering the Taylor series of the error probability at $\sigma = 0$.

**Theorem 3.6.** *Assume that* $\lim_{\sigma \to 0^+} |f^{(i)}_{\mathbf{W}_\mathcal{I}}(\sigma \mathbf{w})| < \infty$, *for all* $\mathcal{I} \subseteq [1:n-1]$ *where* $f^{(i)}_{\mathbf{W}_\mathcal{I}}(\sigma \mathbf{w}) := \frac{\partial^i}{\partial \sigma^i} f_{\mathbf{W}_\mathcal{I}}(\sigma \mathbf{w})$. *Then, the Taylor series of* $P_e(\phi_{\mathrm{lin}}, \mathcal{K})$ *is given by*

$$P_e(\phi_{\mathrm{lin}}, \mathcal{K}) = \sum_{i=0}^{\infty} \frac{P_e^{(i)}}{i!} \sigma^i, \tag{17}$$

*where*

$$P_e^{(i)} = \sum_{k=1}^{\min\{i,n-1\}} (-1)^{k-1} \binom{i}{k} k! \alpha_k^{(i-k)}(0^+),$$

*and*

$$\alpha_k^{(i-k)}(\omega) = \sum_{\substack{\mathcal{I} \subseteq [1:n-1] \\ |\mathcal{I}| = k}} \int_{\mathbf{u} \in \mathbb{R}_+^k} F_{\mathbf{V}_\mathcal{I}}(-\mathbf{u}) f^{(i-k)}_{\mathbf{W}_\mathcal{I}}(\omega \mathbf{u}) \mathrm{d}\mathbf{u},$$

*where* $F_{\mathbf{V}_\mathcal{I}}(\cdot)$ *is the cumulative distribution function (CDF) of* $\mathbf{V}_\mathcal{I}$ *with* $V_i = N_{i+1} - N_i$ *for* $i \in [1:n-1]$.

We defer the proof of Theorem 3.6 to Appendix C. Note that Theorem 3.6 (and also the following Corollary 3.7) generalizes the result in Jeong et al. (2021) under two aspects: (i) from the first-order coefficient to an arbitrary order coefficient; and (ii) beyond Gaussian noise.

As an application of Theorem 3.6, we next present a corollary (proof in Appendix D), which provides the second-order approximation of $P_e(\phi_{\mathrm{lin}}, \mathcal{K})$ for $\mathbf{N} \sim \mathcal{N}(\mathbf{0}_n, \sigma^2 I_n)$ and any exchangeable distribution of $\mathbf{X}$.

**Corollary 3.7.** *Let* $\mathbf{N} \sim \mathcal{N}(\mathbf{0}_n, \sigma^2 I_n)$. *Assume that, for* $i, j \in [1:n-1]$, $|f'_{W_i}(w)| < \infty$, $\forall w$ *and* $f_{W_i, W_j}(u, v) < \infty$, $\forall (u, v)$. *Then, a second order approximation of* $P_e$ *in the low-noise regime is given by*

$$P_e(\phi_{\mathrm{lin}}, \mathcal{K}_\mathbf{N}) = c_1 \sigma + c_2 \sigma^2 + O(\sigma^3), \tag{18}$$

*where* [3]

$$c_1 = \sum_{i=1}^{n-1} \frac{f_{W_i}(0^+)}{\sqrt{\pi}},$$

$$c_2 \approx \frac{1}{2} \sum_{i=1}^{n-1} f'_{W_i}(0^+) - 0.108998 \sum_{i=1}^{n-2} f_{W_i, W_{i+1}}(\mathbf{0}_2^+) - \frac{1}{\pi} \sum_{\substack{(i,j) \in [1:n-1]^2 \\ j > i+1}} f_{W_i, W_j}(\mathbf{0}_2^+).$$

We note that the constants $c_1$ and $c_2$ in Corollary 3.7 depend on the distribution of $\mathbf{X}$. Next, as an example, we derive closed-form expressions for $c_1$ and $c_2$ for $X_i \sim \mathrm{Unif}(0,1)$ and $X_i \sim \mathrm{Exp}(\lambda)$. The detailed proof of these examples can be found in Appendix E, where we also provide various simulation results that graphically showcase the accuracy of the result in Corollary 3.7.

*Example* 3.8. Let $X_i \sim \mathrm{Unif}(0,1)$ and $\mathbf{N} \sim \mathcal{N}(\mathbf{0}_n, \sigma^2 I_n)$. Then, the constants $c_1$ and $c_2$ in Corollary 3.7 are

$$c_1 = \frac{n(n-1)}{\sqrt{\pi}},$$

$$c_2 \approx -\frac{1}{2} n(n-1)^2 - 0.108998 n(n-1)(n-2) - \frac{1}{2\pi} n(n-1)(n-2)(n-3).$$

*Example* 3.9. Let $X_i \sim \mathrm{Exp}(\lambda)$ and $\mathbf{N} \sim \mathcal{N}(\mathbf{0}_n, \sigma^2 I_n)$. Then, the constants $c_1$ and $c_2$ in Corollary 3.7 are

$$c_1 = \frac{n(n-1)\lambda}{2\sqrt{\pi}},$$

$$c_2 \approx -\frac{\lambda^2 n(2n^2 - 3n + 1)}{12} - 0.108998 \frac{\lambda^2 n(n-1)(n-2)}{3} - \frac{\lambda^2 n(n-1)(n-2)(n-3)}{8\pi}.$$

*Remark* 3.10. If $\mathbf{X}$ is exchangeable and $\mathbf{N} \in \mathcal{S}_{n,p}$, $p \geq 1$ in Theorem 3.6, then $P_e(\phi_{\mathrm{lin}}, \mathcal{K}) = P_e(\phi_{\mathrm{opt}}, \mathcal{K})$. This follows since under these conditions, from Theorem 3.3 we have $\phi_{\mathrm{opt}}(\cdot) = \phi_{\mathrm{lin}}(\cdot)$.

---

[3]The approximation of $c_2$ can be made exact by replacing 0.108998 with its exact value $\mathbb{E}[\max\{0, V_1\} \max\{0, V_2\}]$ where $V_i$'s are defined in Thereom 3.6.

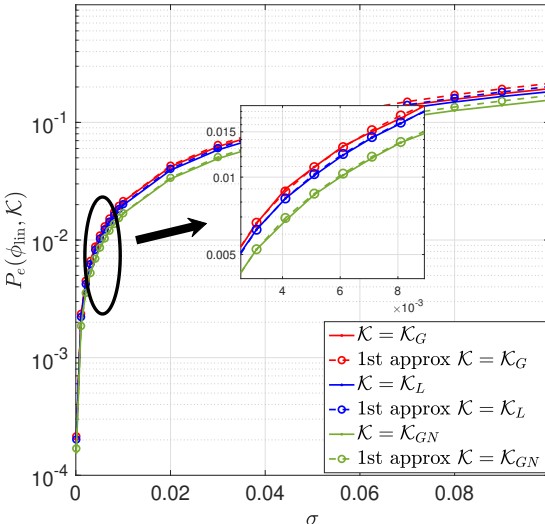

Figure 2: $P_e(\phi_{\text{lin}}, \mathcal{K})$ vs. its first-order approximation.

### 3.3 First-Order Approximation for $P_e$

From Section 3.2, one can infer that the private ranking recovery problem is noise dominated, i.e., the error probability is large even when $\sigma$ is small. For instance, Example 3.8 and Example 3.9 suggest that the first-order coefficient $c_1$ grows quadratically with $n$. Thus, it becomes important to analyze the problem in the low-noise regime, where a reliable permutation recovery can be possible (i.e., $P_e \ll 1$). Towards this end, we next derive the first-order expansion of $P_e(\phi_{\text{lin}}, \mathcal{K})$ with respect to $\sigma$ for any exchangeable $\mathbf{N}$ (note that Corollary 3.7 assumed $\mathbf{N} \sim \mathcal{N}(\mathbf{0}_n, \sigma^2 I_n)$). The proof of the corollary below can be found in Appendix F.

**Corollary 3.11.** *Let $\mathbf{N}$ be exchangeable and $V = N_1 - N_2$. Assume that $f_{W_i}(w) < \infty$, $\forall w$. Then, in the low-noise regime, the first-order approximation of $P_e$ is given by*

$$P_e(\phi_{\text{lin}}, \mathcal{K}) = \frac{C_{\mathbf{X}} \mathbb{E}\left[|V|\right]}{2} \sigma + O(\sigma^2), \tag{19}$$

*with*

$$C_{\mathbf{X}} = \sum_{i=1}^{n-1} f_{W_i}(0^+). \tag{20}$$

*Remark* 3.12. The first-order approximation of $P_e(\phi_{\text{lin}}, \mathcal{K})$ in (19) decouples the effects of the input data distribution (captured by $C_{\mathbf{X}}$) and of the noise distribution (captured by $\mathbb{E}[|V|]$). The assumptions of Corollary 3.11 are not too restrictive: as shown in Jeong et al. (2021), $f_{W_i}(\cdot)$ is bounded if $\mathbf{X}$ is i.i.d., and the PDF of $X$ is bounded.

*Remark* 3.13. For the expansion of $P_e(\phi_{\text{lin}}, \mathcal{K})$ in (19), a natural question arises: How accurate is this? Figure 2 (see more figures in Appendix E.1) shows that this approximation is indeed accurate when $\mathbf{N} \in \mathbb{R}^n$ is i.i.d. according to three different distributions, namely Gaussian (red curve), Laplace (blue curve), and generalized normal with $p = 0.5$ (green curve). Our rationale for choosing such distributions is because in Section 4, we will establish privacy-utility trade-offs for them. In Figure 2, the components of $\mathbf{X}$ were chosen to be i.i.d. according to $\text{Unif}(0, 100)$ with $n = 20$. The solid curves (probability of error) were obtained by Monte-Carlo simulation with $10^6$ iterations, while the dashed curves (first order approximation of the error probability) where obtained by simply evaluating (19).

### 3.4 Input Data Vector with i.i.d. Entries

We here show that, for i.i.d. $X_i \sim X$, the dependence of the first-order approximation of $P_e(\phi_{\mathrm{lin}}, \mathcal{K})$ in (19) on the distribution of $\mathbf{X}$ is rather weak (i.e., it only needs the $L_2$ norm of the PDF of $X$, and not the exact distribution). The approximation of $P_e(\phi_{\mathrm{lin}}, \mathcal{K})$ in (19) depends on the distribution of $\mathbf{X}$ only through $C_{\mathbf{X}}$, and this term can be expressed in closed-form as stated in the following proposition (proof in Appendix G).

**Proposition 3.14.** *Let $\mathbf{X}$ consist of i.i.d. random variables with PDF $f_X(\cdot)$. Then,*

$$C_{\mathbf{X}} = n(n-1)\|f_X\|_2^2, \ \ where \ \|f_X\|_2 = \sqrt{\int_{-\infty}^{\infty} f_X^2(x)\mathrm{d}x}. \tag{21}$$

According to Proposition 3.14 the first-order approximation of $P_e(\phi_{\mathrm{lin}}, \mathcal{K})$ depends on the distribution of $\mathbf{X}$ only through the $L_2$ norm of its PDF. The significance of this result is that we do not need to know the exact distribution of the data vector to analyze $P_e(\phi_{\mathrm{lin}}, \mathcal{K})$.

*Remark* 3.15. Proposition 3.14 shows that $C_{\mathbf{X}}$ grows quadratically in $n$. We now provide a few evaluations of $C_{\mathbf{X}}$ in (21):

- If $X \sim \mathrm{Unif}(a, b)$, $C_{\mathbf{X}} = \frac{n(n-1)}{b-a}$;

- If $X \sim \mathrm{Exp}(\lambda)$, $C_{\mathbf{X}} = \frac{\lambda n(n-1)}{2}$;

- If $X \sim \mathcal{N}(0, 1)$, $C_{\mathbf{X}} = \frac{n(n-1)}{2\sqrt{\pi}}$.

## 4 Privacy and Utility Trade-off

In this section, we investigate the relationship between privacy (measured by the $(\alpha, \epsilon)$-RDP in Definition 2.2) and utility measured by $P_e(\phi_{\mathrm{lin}}, \mathcal{K})$. In particular, we focus on the low-noise regime where, as highlighted in Section 3.3, a reliable permutation recovery is possible.

For a proper definition of DP, we need to consider "well-behaved" query functions (Dwork, 2008). This is the so-called sensitivity property which, for a query function $q(\cdot)$, requires that the sensitivity (formally defined below) is finite.

**Definition 4.1** ($\ell_p$ sensitivity (Liu, 2018)). For all $(\mathbf{X}, \tilde{\mathbf{X}}) \in \mathcal{X}^2$ such that $d_H(\mathbf{X}, \tilde{\mathbf{X}}) \leq 1$, the $\ell_p$ sensitivity of a query $q$ is defined as

$$\Delta_p(q) = \max_{(\mathbf{X}, \tilde{\mathbf{X}}) \in \mathcal{X}^2 : d_H(\mathbf{X}, \tilde{\mathbf{X}}) \leq 1} \|q(\mathbf{X}) - q(\tilde{\mathbf{X}})\|_p, \tag{22}$$

where $p > 0$.

The $\ell_p$ sensitivity is a generalized version of the $\ell_1$ sensitivity for the Laplace mechanism (Dwork et al., 2006b) and of the $\ell_2$ sensitivity for the Gaussian mechanism (Dwork & Roth, 2014). In our framework, we have that the query function $q(\cdot)$ is the identity function, i.e., $q(\mathbf{x}) = \mathbf{x}$. Thus, in order to have a finite $\ell_p$ sensitivity in (22), we need to have a domain constraint on the data input, namely $\mathbf{X} \in \mathcal{X}$ where $\mathcal{X} = \{\mathbf{x} : \mathbf{x} \in [0, \ell]^n\}$. With this, from (22), we have that the $\ell_p$ sensitivity is given by

$$\Delta_p(q) = \Delta(\mathbf{X}) = \max_{(\mathbf{X}, \tilde{\mathbf{X}}) \in \mathcal{X}^2 : d_H(\mathbf{X}, \tilde{\mathbf{X}}) \leq 1} \|\mathbf{X} - \tilde{\mathbf{X}}\|_p = \ell, \ \forall p > 0, \tag{23}$$

and is finite. In what follows, we let $\Delta(\mathbf{X}) = \ell$ denote the $\ell_p$ sensitivity for any $p > 0$, i.e., this notation indicates that the $\ell_p$ sensitivity is independent of $p > 0$. Next, in Section 4.1, we derive a general expression for the privacy-utility trade-off, which holds for any additive noise mechanism. Then, we evaluate it for practically relevant additive noise mechanisms, such as the Laplace (Section 4.2), the Gaussian (Section 4.3), and the generalized normal (Section 4.4) mechanisms.

### 4.1 On the General Trade-off

For the additive noise mechanism in (2), given $(\alpha, \epsilon)$ and the sensitivity $\Delta(\mathbf{X}) = \ell$ in (23), we define the following operation,

$$\mathsf{RDP}_\alpha^{-1}(\epsilon, \ell) = \inf\{\sigma : \mathsf{RDP}_\alpha\left(\mathcal{K}(\sigma, \mathbf{X})\right) \le \epsilon, \Delta(\mathbf{X}) = \ell\}. \tag{24}$$

In words, $\mathsf{RDP}_\alpha^{-1}(\epsilon, \ell)$ is the smallest standard deviation of $\mathcal{K}(\cdot, \cdot)$ that ensures that we meet the $(\alpha, \epsilon)$-RDP constraint when the query sensitivity is equal to $\ell$. If the set in (24) is empty, then we set $\mathsf{RDP}_\alpha^{-1}(\epsilon, \ell) = \infty$.

With the definition in (24) in mind and by using the first-order expansion of $P_e(\phi_{\mathrm{lin}}, \mathcal{K})$ in Corollary 3.11, we arrive at the following general privacy-utility trade-off.

**Proposition 4.2.** *Consider an additive noise mechanism $\mathcal{K}(\sigma, \mathbf{X})$ as in (2) that adopts $\mathbf{N} \in \mathcal{S}_{n,p}$. Let the assumptions in Corollary 3.11 hold. Then, the privacy-utility trade-off for the ranking recovery problem is given by*

$$P_e(\phi_{\mathrm{lin}}, \mathcal{K}) = \frac{\mathbb{E}[|V|]C_\mathbf{X}}{2}\mathsf{RDP}_\alpha^{-1}(\epsilon, \ell) + O\left(\left(\mathsf{RDP}_\alpha^{-1}(\epsilon, \ell)\right)^2\right), \tag{25}$$

*where $V$ and $C_\mathbf{X}$ are defined in Corollary 3.11.*

*Remark* 4.3. The Taylor series of the error probability in Theorem 3.6 allows to characterize higher order approximations for $P_e(\phi_{\mathrm{lin}}, \mathcal{K})$, which in principle can lead to more accurate trade-offs in (25). The error term $O((\mathsf{RDP}_\alpha^{-1}(\epsilon, \ell))^2)$ in (25) arises from the approximation error in the Taylor expansion of $P_e$.

The expression in (25) can, in principle, be used to find a privacy-utility trade-off for *any* additive noise mechanism. As expected, from (25) we note that $P_e(\phi_{\mathrm{lin}}, \mathcal{K})$: (i) decreases as $\epsilon$ increases, and (ii) increases with the data size $n$. Moreover, for i.i.d. data, by using the closed-form expression in (21), we obtain the following trade-off,

$$P_e(\phi_{\mathrm{lin}}, \mathcal{K}) = \frac{n(n-1)\mathbb{E}\left[|V|\right]\|f_X\|_2^2}{2}\mathsf{RDP}_\alpha^{-1}(\epsilon, \ell) + O\left(\left(\mathsf{RDP}_\alpha^{-1}(\epsilon, \ell)\right)^2\right). \tag{26}$$

In the rest of this section, we seek to evaluate Proposition 4.2 and provide results in terms of $\epsilon$ instead of the implicit function $\mathsf{RDP}_\alpha^{-1}(\epsilon, \ell)$. Towards this end, we consider several important mechanisms for which the behavior of $\mathsf{RDP}_\alpha^{-1}(\epsilon, \ell)$ can be determined as a function of $\epsilon$ and $\ell$. For some mechanisms, the expression for $\mathsf{RDP}_\alpha^{-1}(\epsilon, \ell)$ is already known and simply needs to be remapped to our notation (e.g., Gaussian mechanism). For other mechanisms, the expression for $\epsilon$ exists in closed-form, but the inverse $\mathsf{RDP}_\alpha^{-1}(\epsilon, \ell)$ does not have a closed-form (e.g., Laplace mechanism). In such a case, we provide upper and lower bounds on $\mathsf{RDP}_\alpha^{-1}(\epsilon, \ell)$ that indicate its behavior. Yet, in other cases, we find new expressions for $\mathsf{RDP}_\alpha^{-1}(\epsilon, \ell)$ (e.g., generalized normal mechanisms for $\alpha = \infty$).

### 4.2 Laplace Mechanism

We consider a randomized mechanism $\mathcal{K}_L(\sigma, \mathbf{X})$ that consists of adding Laplace noise. Such a mechanism gives $(\alpha, \epsilon)$-RDP as shown in the next result, the proof of which uses the results by Gil et al. (2013) (proof in Appendix H).

**Proposition 4.4.** *For $\alpha > 1$, the randomized mechanism $\mathcal{K}_L(\sigma, \mathbf{X})$ in (2) with $\mathbf{N}$ being i.i.d. according to $Lap(0, b)$ gives $(\alpha, \epsilon)$-RDP with $\epsilon$ given by*

$$\epsilon = \frac{1}{\alpha - 1} \ln \frac{\alpha e^{-(1-\alpha)\ell/(\sigma b)} - (1-\alpha)e^{-\alpha\ell/(\sigma b)}}{2\alpha - 1}. \tag{27}$$

*Moreover, letting $c_\alpha = \frac{1}{\alpha - 1} \ln \frac{\alpha}{2\alpha - 1}$, we have that*

$$\frac{\ell}{\sigma b} + c_\alpha \le \epsilon \le \frac{\ell}{\sigma b} + c_\alpha + \frac{1}{\alpha}e^{-\frac{(2\alpha-1)\ell}{\sigma b}}. \tag{28}$$

We note that Proposition 4.4 is a generalization of the RDP analysis for the Laplace mechanism in (Mironov, 2017) that considered the 1-dimensional case. Although the generalization under the i.i.d. assumption is

straightforward and follows a similar proof, we here reported the proof of Proposition 4.4 for completeness. In addition, the upper and lower bounds on $\epsilon$ are also provided, which we leverage next to provide the privacy-utility trade-off. Furthremore, the gap (i.e., difference) between the upper bound and the lower bound in (28) is given by $\frac{1}{\alpha}e^{-\frac{(2\alpha-1)\ell}{\sigma b}}$. Thus, we can conclude that the bounds in (28) are moderately tight when $\alpha$ is not too small, and the bounds become tight as $\alpha \to \infty$.

We now combine Proposition 4.4 and Corollary 4.2 and obtain an explicit first-order approximation of $P_e$ in terms of $\epsilon$ and $\ell^4$ for the Laplace mechanism in the following corollary (proof in Appendix I).

**Corollary 4.5.** *Let $\mathcal{K}_L(\sigma, \mathbf{X})$ be such that $\mathbf{N}$ is i.i.d. according to $Lap\left(0, \frac{1}{\sqrt{2}}\right)$. Let the assumptions in Corollary 3.11 hold. Then, for $\alpha > 1$, the privacy-utility trade-off is given by*

$$P_e(\phi_{\text{lin}}, \mathcal{K}_L) = \frac{3C_\mathbf{X}}{4\sqrt{2}}\mathsf{RDP}_\alpha^{-1}(\epsilon, \ell) + O\left(\frac{1}{\epsilon^2}\right), \tag{29}$$

*where*

$$\frac{\sqrt{2}\ell}{\left(\epsilon + \frac{1}{\alpha-1}\ln\frac{2\alpha-1}{\alpha}\right)} \leq \mathsf{RDP}_\alpha^{-1}(\epsilon, \ell) \leq \frac{\sqrt{2}\ell}{\epsilon}. \tag{30}$$

Although Corollary 4.5 provides the trade-off in terms of upper and lower bounds, it implies that the trade-off is at least $P_e \propto \frac{1}{\epsilon}$ by considering the lower bound on $\mathsf{RDP}_\alpha^{-1}(\epsilon, \ell)$. We note that for the case of $\alpha = \infty$, which is equivalent to $\epsilon$-DP, the bound in (30) becomes exact and $\mathsf{RDP}_\alpha^{-1}(\epsilon, \ell) = \frac{\sqrt{2}\ell}{\epsilon}$.

*Remark* 4.6. Since $\mathsf{RDP}_\alpha^{-1}(\epsilon, \ell)$ does not have a closed-form, the bounds on $\mathsf{RDP}_\alpha^{-1}(\epsilon, \ell)$ in (30) were provided to indicate its behavior with respect to $\epsilon$ and $\ell$. However, if one needs an exact value of $\mathsf{RDP}_\alpha^{-1}(\epsilon, \ell)$ for a given $(\epsilon, \ell)$, this can easily be done numerically by inverting (27).

As an example, we next evaluate (29) when $\mathbf{X}$ is i.i.d. and has a uniform distribution.

*Example* 4.7. If $\mathbf{X} \sim \text{Unif}([0, \ell]^n)$, then $C_\mathbf{X} = \frac{n(n-1)}{\ell}$ and the trade-off in Corollary 4.5 becomes

$$P_e(\phi_{\text{lin}}, \mathcal{K}_L) = \frac{3n(n-1)}{4}\mathsf{R}_\alpha^{-1}(\epsilon) + O\left(\frac{1}{\epsilon^2}\right),$$

where

$$\frac{1}{\epsilon + \frac{1}{\alpha-1}\ln\frac{2\alpha-1}{\alpha}} \leq \mathsf{R}_\alpha^{-1}(\epsilon) \leq \frac{1}{\epsilon}.$$

### 4.3 Gaussian Mechanism

We here analyze a mechanism $\mathcal{K}_G(\sigma, \mathbf{X})$ that consists of adding Gaussian noise. This gives $(\alpha, \epsilon)$-RDP as shown in the next result, the proof of which can be found in Appendix J and uses the results in Gil et al. (2013). We note that this result was already derived by Mironov (2017), but we report it here for completeness.

**Proposition 4.8.** *$\mathcal{K}_G(\sigma, \mathbf{X})$ in (2) with $\mathbf{N}$ being i.i.d. according to $\mathcal{N}(0, 1)$ gives $\left(\alpha, \frac{\alpha\ell^2}{2\sigma^2}\right)$-RDP. Consequently,*

$$\mathsf{RDP}_\alpha^{-1}(\epsilon, \ell) = \sqrt{\frac{\alpha\ell^2}{2\epsilon}}. \tag{31}$$

We now evaluate the trade-off stated in Proposition 4.2. For independent standard Gaussian random variables $N_1$ and $N_2$, we have that $\mathbb{E}[|V|] = \mathbb{E}[|N_1 - N_2|] = \frac{2}{\sqrt{\pi}}$. By leveraging Proposition 4.8 and Corollary 3.11, we then obtain the privacy-utility trade-off for the Gaussian mechanism as shown in the following corollary.

**Corollary 4.9.** *Consider the Gaussian mechanism $\mathcal{K}_G(\sigma, \mathbf{X})$ with $\mathbf{N}$ being i.i.d. according to $\mathcal{N}(0, 1)$. Let the assumptions in Corollary 3.11 hold. Then, for $\alpha \geq 1$, the privacy-utility trade-off is given by*

$$P_e(\phi_{\text{lin}}, \mathcal{K}_G) = \frac{\ell C_\mathbf{X}}{\sqrt{2\pi}}\sqrt{\frac{\alpha}{\epsilon}} + O\left(\frac{1}{\epsilon}\right). \tag{32}$$

---

[4]The approximation arises from the Taylor series of $P_e$ in Corollary 3.11.

From (32) we observe that the Gaussian mechanism gives a $P_e$ that is inversely proportional to $\sqrt{\epsilon}$, while the Laplace mechanism in (29) offers a $P_e$ that scales inversely proportional to $\epsilon$ as shown in Corollary 4.5. Thus, we can conclude that the Laplace mechanism outperforms the Gaussian mechanism in terms of the rate of the privacy-utility trade-off. We complete this subsection by giving an example when $\mathbf{X}$ is i.i.d. and has a uniform distribution.

*Example* 4.10. If $\mathbf{X} \sim \text{Unif}([0, \ell]^n)$, then $C_{\mathbf{X}} = \frac{n(n-1)}{\ell}$ and the trade-off in Corollary 4.9 becomes

$$P_e(\phi_{\text{lin}}, \mathcal{K}_G) = \frac{n(n-1)}{\sqrt{2\pi}}\sqrt{\frac{\alpha}{\epsilon}} + O\left(\frac{1}{\epsilon}\right).$$

### 4.4 Generalized Normal Mechanism

Corollary 4.5 and Corollary 4.9 suggest that $P_e \propto (1/\epsilon)^{1/p}$, where $p$ is the power of the exponent in the noise PDF. In other words, the smaller the $p$ is, the better the trade-off appears to be. Motivated by this observation, we consider a generalized normal mechanism (Liu, 2018) denoted by $\mathcal{K}_{GN}$ where $\mathbf{N}$ is i.i.d. according to $\mathcal{GN}(0, a, p)$ with $p \leq 1$. Although $p$ can be greater than 1, we only consider $p \leq 1$ as motivated by the trade-off $P_e \propto (1/\epsilon)^{1/p}$. Different from the previous RDP analysis for the Laplace and Gaussian mechanisms, we here study only $\epsilon$-DP for $\mathcal{K}_{GN}$ (i.e., $\alpha = \infty$). Recall that $\epsilon$-DP offers a stronger privacy guarantee than RDP. The $\epsilon$-DP of $\mathcal{K}_{GN}$ is given in the next proposition (proof in Appendix K).

**Proposition 4.11.** *Let $\mathbf{N}$ be i.i.d. according to $N \sim \mathcal{GN}(0, h(p), p)$ with $p \leq 1$ and $h(p) = \sqrt{\frac{\Gamma(p^{-1})}{\Gamma(3p^{-1})}}$, where $\Gamma(\cdot)$ is the gamma function. Then, the generalized normal mechanism $\mathcal{K}_{GN}(\sigma)$ gives $\epsilon$-DP with*

$$\epsilon = \left(\frac{\ell}{\sigma h(p)}\right)^p. \tag{33}$$

*Consequently,*

$$\text{RDP}_\infty^{-1}(\epsilon, \ell) = \frac{\ell}{h(p)}\left(\frac{1}{\epsilon}\right)^{\frac{1}{p}}. \tag{34}$$

*Remark* 4.12. We note that the work of (Liu, 2018) only considered integer values for the parameter $p$. The above result extends the work of (Liu, 2018) to any $p \in (0, 1]$.

We combine Proposition 4.2 and Proposition 4.11 and obtain the trade-off in the corollary below.

**Corollary 4.13.** *Consider the generalized normal mechanism $\mathcal{K}_{GN}(\sigma, \mathbf{X})$ with $p \leq 1$. Let the assumptions in Corollary 3.11 hold. Then, the privacy-utility trade-off is given by*

$$P_e(\phi_{\text{lin}}, \mathcal{K}_{GN}) = \frac{\mathbb{E}[|N - N'|]\ell C_{\mathbf{X}}}{2h(p)}\left(\frac{1}{\epsilon}\right)^{\frac{1}{p}} + O\left(\frac{1}{\epsilon^{\frac{2}{p}}}\right), \tag{35}$$

*where $N$ and $N'$ are independent and $N' \overset{d}{=} N$.*

*Remark* 4.14. Corollary 4.13 confirms our observation that the smaller the $p$ is, the better the trade-off is. Thus, for $\alpha = \infty$ (or $\epsilon$-DP), the generalized normal distribution with $p \leq 1$ offers a better privacy-utility trade-off than the Laplace and Gaussian mechanisms. In addition, note that the constant in the first-order term of (35) can be upper bound by using Jensen's inequality as follows,

$$\mathbb{E}[|N - N'|] \leq \sqrt{\mathbb{E}[|N - N'|^2]} = \sqrt{2\text{Var}(N)} = \sqrt{2},$$

where we have used the fact that $\text{Var}(N) = 1$. Furthermore, we seek to minimize (35) with respect to $p$ in order to find the best generalized normal mechanism given an $\epsilon$-DP constraint. We refer to Appendix L, where we discuss this and provide the best $p$.

## 5    Conclusions

We studied the private ranking recovery problem within the DP framework. We designed a low-complexity decoder and characterized sufficient conditions for its optimality. We derived the Taylor series of the error probability when such a decoder is used, as well as the first-order approximation of it. We leveraged the first-order approximation of the error probability, along with the $(\alpha, \epsilon)$-RDP, to obtain utility-privacy trade-offs for the Gaussian, Laplace, and generalized normal mechanisms. These results allow us to compare different noise mechanisms in order to determine the best utility-privacy trade-off. In addition, our results show that the problem of private ranking recovery is noise dominated, i.e., the error probability is large even for small values of the noise variance. This suggests that the exact recovery imposed in our work might need to be relaxed. Finally, possible future directions include the following: (i) *partial recovery* in which we seek to recover the permutation of only part of the input data; (ii) *approximate recovery* in which we allow a fixed number of errors given a ranking distance function (Kumar & Vassilvitskii, 2010) (e.g., Hamming distance, Kendall's tau distance); (iii) investigating or generalizing the results in this paper to hold universally, for any distribution on the input data vector.

### Acknowledgments

The work of M. Jeong and M. Cardone was supported in part by the U.S. National Science Foundation under Grant CCF-1849757. The authors are grateful to the reviewers and the editor for useful comments and suggestions.

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

# A  Proof of Example 2.5

## A.1  Gaussian Noise

If $\mathbf{N} \sim \mathcal{N}(\mathbf{0}_n, \sigma^2 I_n)$, its PDF is

$$f_{\mathbf{N}}(\mathbf{z}) = \frac{1}{(2\pi)^{n/2}\sigma^n} e^{-\frac{\|\mathbf{z}\|_2^2}{2\sigma^2}} = g(\|\mathbf{z}\|_2), \tag{36}$$

with $g(t) = \frac{1}{(2\pi)^{n/2}\sigma^n} e^{-\frac{t^2}{2\sigma^2}}$. Since $g(t)$ is a non-increasing function in $t > 0$, then $\mathcal{N}(\mathbf{0}_n, \sigma^2 I_n) \in \mathcal{S}_{n,2}$.

## A.2  Laplace Noise

If $\mathbf{N}$ consists of i.i.d. $N_i \sim \mathrm{Lap}(0, b)$, its PDF is

$$f_{\mathbf{N}}(\mathbf{z}) = \prod_{i=1}^{n} \frac{1}{2b} e^{-\frac{|z_i|}{b}} = \frac{1}{(2b)^n} e^{-\frac{\|\mathbf{z}\|_1}{b}} = g(\|\mathbf{z}\|_1), \tag{37}$$

with $g(t) = \frac{1}{(2b)^n} e^{-\frac{t}{b}}$, which is a non-increasing function in $t > 0$. Thus, a joint distribution of i.i.d. $\mathrm{Lap}(0, b)$ is a member of $\mathcal{S}_{n,1}$.

## A.3  Generalized Normal Noise

If $\mathbf{N}$ consists of i.i.d. $N_i \sim \mathcal{GN}(0, a, p)$, its PDF is Nadarajah (2005)

$$f_{\mathbf{N}}(\mathbf{z}) = \prod_{i=1}^{n} K \exp\left(-\left|\frac{z_i}{a}\right|^p\right) = K^n \exp\left(-\frac{\sum_{i=1}^{n} |z_i|^p}{a^p}\right) = g(\|\mathbf{z}\|_p), \tag{38}$$

where $K = \frac{p}{2a\Gamma(1/p)}$ is the normalization factor, and $g(t) = K^n \exp\left(-\frac{t^p}{a^p}\right)$, which is a non-increasing function in $t > 0$. Thus, a joint distribution of i.i.d. $\mathcal{GN}(0, a, p)$ is a member of $\mathcal{S}_{n,p}$.

### A.4 Staircase Noise

If $\mathbf{N}$ has a staircase distribution with $(\lambda, \gamma, \Delta)$, its PDF is of the form Geng et al. (2015)

$$f_{\mathbf{N}}(\mathbf{z}) = \beta e^{-\lambda h(\mathbf{z})}, \tag{39}$$

with

$$h(\mathbf{z}) = \begin{cases} k & \text{if } \|\mathbf{z}\|_1 \in [k\Delta, (k+\gamma)\Delta], \\ k+1 & \text{if } \|\mathbf{z}\|_1 \in [(k+\gamma)\Delta, (k+1)\Delta], \end{cases} \tag{40}$$

where $\gamma \in [0, 1]$ and $\Delta > 0$ are given, and $\beta$ is a normalization parameter.

Since $h(\mathbf{z})$ is a non-decreasing function in $\|\mathbf{z}\|_1$, then $f_{\mathbf{N}}(\mathbf{z})$ is non-increasing in $\|\mathbf{z}\|_1$. Thus, a staircase distribution is a member of $\mathcal{S}_{n,1}$.

### A.5 Uniform Noise

If $\mathbf{N} \sim \text{Unif}(\mathcal{B}_p(\mathbf{0}_n, r))$ with $r > 0$, its PDF is given by

$$\begin{aligned} f_{\mathbf{N}}(\mathbf{z}) &= \frac{1}{\text{Vol}(\mathcal{B}_p(\mathbf{0}_n, r))} \mathbb{1}_{\{\mathbf{z} \in \mathcal{B}_p(\mathbf{0}_n, r)\}} \\ &= \frac{1}{\text{Vol}(\mathcal{B}_p(\mathbf{0}_n, r))} \mathbb{1}_{\{\|\mathbf{z}\|_p \leq r\}}, \end{aligned} \tag{41}$$

where $\text{Vol}(\mathcal{S})$ denotes the volume of the set $\mathcal{S}$. Clearly, the PDF in (41) is a non-increasing function in $\|\mathbf{z}\|_p$ and hence, $\text{Unif}(\mathcal{B}_p(\mathbf{0}_n, r))$ is $\mathcal{S}_{n,p}$.

## B Proof of Lemma 3.2

Our goal is to show that a solution $\hat{\kappa}$ for the following optimization problem,

$$\arg\min_{\kappa \in \mathcal{P}} \|\mathbf{y} - P_{\eta \to \kappa}\mathbf{x}\|_p, \tag{42}$$

for any $\eta \in \mathcal{P}$, $\mathbf{y} \in \mathcal{H}_\tau$ and $\mathbf{x} \in \mathcal{H}_\eta$, is given by $\hat{\kappa} = \tau$. We start by noting that, by the property of permutation invariance of the $\ell_p$-norm, without loss of generality, we can consider $\tau = (1, 2, \ldots, n)$ which indicates that $\mathbf{y}$ is sorted in ascending order. In addition, a solution to (42) does not depend on the permutation of $\mathbf{x}$, i.e., we can start the problem with any $\mathbf{x} \in \mathcal{H}_\eta$. This is beause we consider every possible permutation matrix $P_{\eta \to \kappa}$'s with $\kappa \in \mathcal{P}$. Hence, we set $\eta = \tau$, which implies that $\mathbf{x}$ is also sorted according to the ascending order, i.e., $\mathbf{x}$ and $\mathbf{y}$ are sorted according to the same permutation $\tau$.

The key tool that will enable our proof is the following generalized version of the rearrangement inequality Holstermann (2017).

**Lemma B.1.** *Consider a sequence of real numbers $a_1 \leq \ldots \leq a_n$ and a collection of functions $f_i(\cdot) : [a_1, a_n] \mapsto \mathbb{R}$ for $i \in [1 : n]$ and for some fixed $n$. Suppose that for all $x \in [a_1, a_n]$ we have that*

$$f_1'(x) \leq \ldots \leq f_n'(x). \tag{43}$$

*Then, for any permutation $\kappa \in \mathcal{P}$, we have that*

$$\sum_{i=1}^{n} f_i(a_{n-i+1}) \leq \sum_{i=1}^{n} f_i(a_{\kappa_{n-i+1}}), \tag{44}$$

*where $a_{\kappa_i} = (P_{\tau \to \kappa}\mathbf{a})_i$, $i \in [1 : n]$.*

For the given $\mathbf{y} \in \mathcal{H}_\tau$, in order to apply Lemma B.1, we define a sequence of functions

$$f_i(t) \triangleq |y_{n-i+1} - t|^p, \ i \in [1:n]. \tag{45}$$

Note that $y_1 \le y_2 \le \cdots \le y_n$. For the time being assume that

$$f_i'(t) \le f_j'(t), \ \forall t \in \mathbb{R}, \tag{46}$$

for all $i < j$. The claim in (46), which will be shown later, guarantees that we can use Lemma B.1. Therefore, by setting $a_i = x_i$ in Lemma B.1, and recalling that $\mathbf{x}$ and $\mathbf{y}$ are sorted according to the same permutation $\tau$, we arrive at

$$\begin{aligned}
\|\mathbf{y} - \mathbf{x}\|_p^p &= \sum_{i=1}^n |y_{n-i+1} - x_{n-i+1}|^p \\
&= \sum_{i=1}^n f_i(x_{n-i+1}) \\
&\le \sum_{i=1}^n f_i(x_{\kappa_{n-i+1}}) \\
&= \sum_{i=1}^n |y_{n-i+1} - x_{\kappa_{n-i+1}}|^p \\
&= \sum_{i=1}^n |y_{n-i+1} - (P_{\tau \to \kappa}\mathbf{x})_{n-i+1}|^p \\
&= \|\mathbf{y} - P_{\tau \to \kappa}\mathbf{x}\|_p^p,
\end{aligned}$$

for all $\kappa \in \mathcal{P}$. This indeed shows that, under the assumption in (46), a solution $\hat{\kappa}$ for the optimization problem in (42) is given by $\hat{\kappa} = \tau$.

To complete the proof it remains to verify that the condition in (46) holds. Towards this end, we observe that

$$f_i'(t) = p(t - y_{n-i+1})|t - y_{n-i+1}|^{p-2},$$

for all $i \in [1:n]$. We now show that $f_i'(t) \le f_j'(t)$ for all $t \in \mathbb{R}$ and $i < j$. This follows by a simple comparison, which consists of subtracting $f_j'(t)$ from $f_i'(t)$, namely

$$f_i'(t) - f_j'(t) = p(t - y_{n-i+1})|t - y_{n-i+1}|^{p-2} - p(t - y_{n-j+1})|t - y_{n-j+1}|^{p-2}, \tag{47}$$

where $y_{n-i+1} \ge y_{n-j+1}$ since $\mathbf{y}$ by assumption is sorted in ascending order. If (47) is less than or equal to zero, then (46) holds. We now show that (47) is indeed always less than or equal to zero.

- $t \in [-\infty, y_{n-j+1}]$: In this case, (47) becomes

$$f_i'(t) - f_j'(t) = -p(y_{n-i+1} - t)^{p-1} + p(y_{n-j+1} - t)^{p-1},$$

  which is always less than or equal to zero;

- $t \in [y_{n-j+1}, y_{n-i+1}]$: In this case, (47) becomes

$$f_i'(t) - f_j'(t) = -p(y_{n-i+1} - t)^{p-1} - p(t - y_{n-j+1})^{p-1},$$

  which is always less than or equal to zero;

- $t \in [y_{n-i+1}, \infty]$: In this case, (47) becomes

$$f_i'(t) - f_j'(t) = -p(t - y_{n-i+1})^{p-1} - p(t - y_{n-j+1})^{p-1},$$

  which is always less than or equal to zero.

The above three cases imply that the inequality in (46) holds for any $i < j$ and hence, the sequence of functions in (45) satisfies (43). This concludes the proof of the desired claim and the proof of Lemma 3.2.

## C   Proof of Theorem 3.6

The probability of error is given by Jeong et al. (2021, Corollary 1),

$$P_e(\phi_{\text{lin}}, \mathcal{K}) = 1 - \mathbb{E}\left[\Pr\left(\mathbf{V} \geq -\frac{T_\tau \mathbf{X}}{\sigma} \,\Big|\, \mathbf{X}\right) \,\Big|\, \mathbf{X} \in \mathcal{H}_\tau\right], \tag{48}$$

where $T_\tau$ is defined in (15). We now note that, from Remark 3.5, we can write $T_\tau \mathbf{X} | \mathbf{X} \in \mathcal{H}_\tau$ as a spacing vector $\mathbf{W}$ and hence, we can equivalently rewrite (48) as

$$P_e(\phi_{\text{lin}}, \mathcal{K}) = 1 - \Pr\left(\bigcap_{i=1}^{n-1}\left\{V_i \geq \frac{-W_i}{\sigma}\right\}\right)$$

$$= \Pr\left(\bigcup_{i=1}^{n-1}\left\{V_i < \frac{-W_i}{\sigma}\right\}\right)$$

$$= \sum_{k=1}^{n-1}\left((-1)^{k-1}\sum_{\substack{\mathcal{I} \subseteq [1:n-1] \\ |\mathcal{I}|=k}} \Pr\left(\mathcal{A}_\mathcal{I}\right)\right), \tag{49}$$

where the last equality follows from the inclusion-exclusion principle where $\mathcal{A}_\mathcal{I} = \cap_{i \in \mathcal{I}}\mathcal{A}_i$ with $\mathcal{A}_i = \{V_i < -\sigma^{-1}W_i\}$.

For any set $|\mathcal{I}| = k$, we have

$$\Pr(\mathcal{A}_\mathcal{I}) = \Pr\left(\bigcap_{i \in \mathcal{I}}\left\{V_i < \frac{-W_i}{\sigma}\right\}\right)$$

$$= \int_{\mathbf{w} \in \mathbb{R}_+^k} F_{\mathbf{V}_\mathcal{I}}\left(\frac{-\mathbf{w}}{\sigma}\right) f_{\mathbf{W}_\mathcal{I}}(\mathbf{w})\mathrm{d}\mathbf{w}$$

$$= \int_{\mathbf{u} \in \mathbb{R}_+^k} F_{\mathbf{V}_\mathcal{I}}(-\mathbf{u}) f_{\mathbf{W}_\mathcal{I}}(\sigma\mathbf{u})\sigma^k\mathrm{d}\mathbf{u}, \tag{50}$$

where the last equality follows from a change of variable. By substituting (50) into (49), we obtain

$$P_e(\phi_{\text{lin}}, \mathcal{K}) = \sum_{k=1}^{n-1}(-1)^{k-1}\alpha_k(\sigma)\sigma^k, \tag{51}$$

where

$$\alpha_k(\sigma) = \sum_{\substack{\mathcal{I} \subseteq [1:n-1] \\ |\mathcal{I}|=k}} \int_{\mathbf{u} \in \mathbb{R}_+^k} F_{\mathbf{V}_\mathcal{I}}(-\mathbf{u}) f_{\mathbf{W}_\mathcal{I}}(\sigma\mathbf{u})\mathrm{d}\mathbf{u}. \tag{52}$$

Let $f_{\mathbf{W}_\mathcal{I}}^{(m)}(\sigma\mathbf{u}) = \frac{\partial^m}{\partial\sigma^m}f_{\mathbf{W}_\mathcal{I}}(\sigma\mathbf{u})$. By the Leibniz integral rule, the $m$-th derivative of $\alpha_k(\sigma)$ w.r.t. $\sigma$ is

$$\alpha_k^{(m)}(\sigma) = \sum_{\substack{\mathcal{I} \subseteq [1:n-1] \\ |\mathcal{I}|=k}} \int_{\mathbf{u} \in \mathbb{R}_+^k} F_{\mathbf{V}_\mathcal{I}}(-\mathbf{u}) f_{\mathbf{W}_\mathcal{I}}^{(m)}(\sigma\mathbf{u})\mathrm{d}\mathbf{u}. \tag{53}$$

Since (53) is bounded at $\sigma \to 0^+$ (i.e., $\lim_{\sigma \to 0^+}|\alpha_k^{(m)}(\sigma)| < \infty$), after some trivial algebra, we obtain that the $m$-th derivative of $P_e(\phi_{\text{lin}}, \mathcal{K})$ in (51) at $\sigma \to 0^+$ is

$$P_e^{(m)}(\sigma)\Big|_{\sigma \to 0^+} = \sum_{k=1}^{\min\{m,n-1\}}(-1)^{k-1}\binom{m}{k}k!\alpha_k^{(m-k)}(0^+). \tag{54}$$

We conclude the proof of Theorem 3.6 by plugging (54) into the Taylor series of $P_e(\phi_{\text{lin}}, \mathcal{K})$ at $\sigma = 0^+$.

## D  Proof of Corollary 3.7

Since $P_e(\phi_{\text{lin}}, \mathcal{K}) \to 0$ as $\sigma \to 0^+$, the first order rate is given from Theorem 3.6 by

$$
\begin{aligned}
P_e^{(1)} &= \lim_{\sigma \to 0^+} \sum_{\substack{\mathcal{I} \subseteq [1:n-1] \\ |\mathcal{I}|=1}} \int_{\mathbf{u} \in \mathbb{R}_+} F_{\mathbf{V}_{\mathcal{I}}}(-\mathbf{u}) f_{\mathbf{W}_{\mathcal{I}}}(\sigma \mathbf{u}) \, \mathrm{d}\mathbf{u} \\
&= \lim_{\sigma \to 0^+} \sum_{i=1}^{n-1} \int_{u \in \mathbb{R}_+} \Pr(V_i \le -u) f_{W_i}(\sigma u) \, \mathrm{d}u \\
&\overset{(a)}{=} \lim_{\sigma \to 0^+} \sum_{i=1}^{n-1} \int_{v \in \mathbb{R}_+} Q(v) f_{W_i}(\sqrt{2}\sigma v) \sqrt{2} \, \mathrm{d}v \\
&\overset{(b)}{=} \sum_{i=1}^{n-1} f_{W_i}(0^+) \int_{v \in \mathbb{R}_+} Q(v) \sqrt{2} \, \mathrm{d}v \\
&= \sum_{i=1}^{n-1} \frac{f_{W_i}(0^+)}{\sqrt{\pi}},
\end{aligned}
\tag{55}
$$

where (a) follows using the change of variable $u = \sqrt{2}v$ toegther with the fact that $V_i \sim \mathcal{N}(0, 2)$ with $Q(\cdot)$ being the Q function of the standard normal distribution; and (b) follows by the dominated convergence theorem.

The second order rate is also given from Theorem 3.6 by

$$
\frac{1}{2} P_e^{(2)} = \frac{1}{2} \sum_{k=1}^{2} (-1)^{k-1} \binom{2}{k} k! \alpha_k^{(2-k)}(0^+) = \alpha_1^{(1)}(0^+) - \alpha_2(0^+).
\tag{56}
$$

We need to compute $\alpha_1^{(1)}(0^+)$ and $\alpha_2(0^+)$. Firstly, we have

$$
\begin{aligned}
\alpha_1^{(1)}(0^+) &= \lim_{\sigma \to 0^+} \sum_{i=1}^{n-1} \int_0^\infty F_{V_i}(-u) f_{W_i}^{(1)}(\sigma u) \, \mathrm{d}u \\
&\overset{(a)}{=} \lim_{\sigma \to 0^+} \sum_{i=1}^{n-1} \int_{u \in \mathbb{R}_+} \Pr(V_i \le -u) u f_{W_i}'(\sigma u) \, \mathrm{d}u \\
&\overset{(b)}{=} \sum_{i=1}^{n-1} f_{W_i}'(0^+) \int_{u \in \mathbb{R}_+} \Pr(V_i \le -u) u \, \mathrm{d}u \\
&\overset{(c)}{=} \sum_{i=1}^{n-1} f_{W_i}'(0^+) \int_{v \in \mathbb{R}_+} Q(v) 2v \, \mathrm{d}v \\
&= \frac{1}{2} \sum_{i=1}^{n-1} f_{W_i}'(0^+),
\end{aligned}
\tag{57}
$$

where the labeled equalities follow from: (a) letting $f_{W_i}'(\sigma u) = \frac{\partial}{\partial w} f_{W_i}(w)\big|_{w = \sigma u}$; (b) using the dominated convergence theorem; and (c) using the change of variable $u = \sqrt{2}v$ similar to the step (a) in (55).

The second term in (56) is given from Theorem 3.6 by

$$
\begin{aligned}
\alpha_2(0^+) &= \lim_{\sigma \to 0^+} \sum_{\substack{\mathcal{I} \subseteq [1:n-1] \\ |\mathcal{I}|=2}} \int_{\mathbf{u} \in \mathbb{R}_+^2} \Pr(\mathbf{V}_{\mathcal{I}} \leq -\mathbf{u}) f_{\mathbf{W}_{\mathcal{I}}}(\sigma \mathbf{u}) \, \mathrm{d}\mathbf{u} \\
&\overset{(a)}{=} \sum_{\substack{\mathcal{I} \subseteq [1:n-1] \\ |\mathcal{I}|=2}} f_{\mathbf{W}_{\mathcal{I}}}(\mathbf{0}_2^+) \int_{\mathbf{u} \in \mathbb{R}_+^2} \Pr(\mathbf{V}_{\mathcal{I}} \leq -\mathbf{u}) \, \mathrm{d}\mathbf{u} \\
&= \sum_{i=1}^{n-2} f_{W_i, W_{i+1}}(\mathbf{0}_2^+) \int_{\mathbf{u} \in \mathbb{R}_+^2} \Pr(V_i \leq -u_1, V_{i+1} \leq -u_2) \, \mathrm{d}\mathbf{u} \\
&\quad + \sum_{\substack{(i,j) \in [1:n-1]^2 \\ j > i+1}} f_{W_i, W_j}(\mathbf{0}_2^+) \int_{\mathbf{u} \in \mathbb{R}_+^2} \Pr(V_i \leq -u_1, V_j \leq -u_2) \, \mathrm{d}\mathbf{u} \\
&\overset{(b)}{=} \sum_{i=1}^{n-2} f_{W_i, W_{i+1}}(\mathbf{0}_2^+) \int_{\mathbf{u} \in \mathbb{R}_+^2} Q_{V_i, V_{i+1}}(\mathbf{u}) \, \mathrm{d}\mathbf{u} + \sum_{\substack{(i,j) \in [1:n-1]^2 \\ j > i+1}} f_{W_i, W_j}(\mathbf{0}_2^+) \left( \int_{u \in \mathbb{R}_+} Q_V(u) \, \mathrm{d}u \right)^2 \\
&= \mathbb{E}[\max\{0, V_1\} \max\{0, V_2\}] \sum_{i=1}^{n-2} f_{W_i, W_{i+1}}(\mathbf{0}_2^+) + \mathbb{E}^2[\max\{0, V_1\}] \sum_{\substack{(i,j) \in [1:n-1]^2 \\ j > i+1}} f_{W_i, W_j}(\mathbf{0}_2^+) \\
&\overset{(c)}{\approx} 0.108998 \sum_{i=1}^{n-2} f_{W_i, W_{i+1}}(\mathbf{0}_2^+) + \frac{1}{\pi} \sum_{\substack{(i,j) \in [1:n-1]^2 \\ j > i+1}} f_{W_i, W_j}(\mathbf{0}_2^+),
\end{aligned} \tag{58}
$$

where (a) follows by the dominated convergence theorem; (b) is due to the independence of $V_i$ and $V_j$ if $|i - j| > 1$ (since $V_i = N_{i+1} - N_i$ and $V_j = N_{j+1} - N_j$ with i.i.d. $\mathbf{N}$); (c) follows by noting that $\mathbb{E}[\max\{0, V_1\} \max\{0, V_2\}] \approx 0.108998$ and $\mathbb{E}^2[\max\{0, V_1\}] = \frac{1}{\pi}$.

By substituting (57) and (58) into (56), we obtain

$$
\frac{1}{2} P_e^{(2)} \approx \frac{1}{2} \sum_{i=1}^{n-1} f'_{W_i}(0^+) - 0.108998 \sum_{i=1}^{n-2} f_{W_i, W_{i+1}}(\mathbf{0}_2^+) - \frac{1}{\pi} \sum_{\substack{(i,j) \in [1:n-1]^2 \\ j > i+1}} f_{W_i, W_j}(\mathbf{0}_2^+). \tag{59}
$$

This concludes the proof of Corollary 3.7.

# E   Simulation Results and Proof of Example 3.8 and 3.9

## E.1   Simulation Results shown in Figure 3

For the simulations illustrated in Figure 3, we set $\mathrm{Unif}(0, 1)$ and $\mathrm{Exp}(1)$ for $X_i, i \in [1 : n]$ and $\mathbf{N} \sim \mathcal{N}(\mathbf{0}_n, \sigma^2 I_n)$. The curves for the true error probability $P_e(\phi_{\mathrm{lin}})$ were obtained by Monte-Carlo simulation using $10^6$ iterations, whereas we obtained the curves for the first and second order approximations by evaluating the expression in Corollary 3.7. The data dimension is set to $n = 10$ for (a) and (b), and to $n = 20$ for (c) and (d). It is shown from (a) and (c) that the first and second-order approximations well fit the true $P_e(\phi_{\mathrm{lin}})$ around $\sigma = 0$. Further, (b) and (d) show the approximations in the low-noise regime and illustrate that, if the targeted error probability is small, then the first-order approximation is very close to the true error probability.

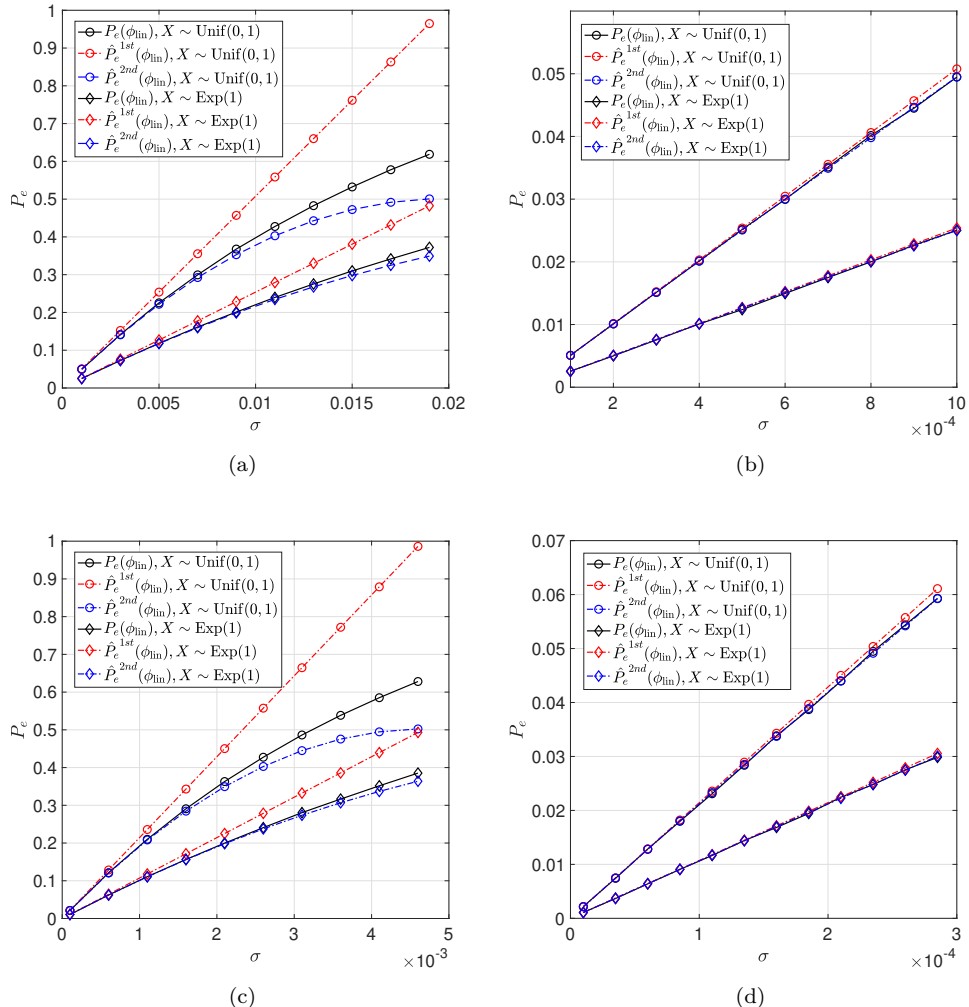

Figure 3: Comparison between $P_e(\phi_{\lin})$, the first-order approximation $\hat{P}_e^{1st}(\phi_{\lin})$, and the second-order approximation $\hat{P}_e^{2nd}(\phi_{\lin})$. We set $X_i \sim \mathrm{Unif}(0,1)$ and $X_i \sim \mathrm{Exp}(1)$ for $i \in [1:n]$: (a) $n = 10$; (b) $n = 10$ in low-noise; (c) $n = 20$; (d) $n = 20$ in low-noise.

## E.2  Proof of Example 3.8

To evaluate the expression in Corollary 3.7, we need $f_{W_i}(0^+)$, $f'_{W_i}(0^+)$ and $f_{W_i,W_j}(\mathbf{0}_2^+)$. For $X_i \sim \mathrm{Unif}(0,1)$, the PDF of the spacing $W_i$ and $W_j, W_i$ are given by Pyke (1965)

$$f_{W_i}(w) = n(1-w)^{n-1}, \ \forall i \in [1:n-1], \tag{60}$$

$$f_{W_i,W_j}(u,v) = n(n-1)(1-u-v)^{n-2}, \ \forall i \neq j, \tag{61}$$

which gives

$$f_{W_i}(0^+) = n, \ \forall i,$$
$$f_{W_i,W_j}(\mathbf{0}_2^+) = n(n-1), \ \forall i \neq j.$$

By differentiating (60) with respect to $w$, we also obtain

$$f'_{W_i}(w) = -n(n-1)(1-w)^{n-2},$$
$$\implies f'_{W_i}(0^+) = -n(n-1), \ \forall i.$$

This concludes the proof of Example 3.8

### E.3 Proof of Example 3.9

To evaluate $c_1$ and $c_2$, we make use of the fact Pyke (1965) that the spacings of $\text{Exp}(\lambda)$ random variables are independent exponential random variables with parameters depending on the dimension $n$ and $\lambda$. Specifically, for i.i.d. $X_i \sim \text{Exp}(\lambda)$ the spacings become independent $W_i$'s that are distributed as

$$W_i \sim \text{Exp}(\lambda(n-i)), \ \forall i \in [1:n-1]. \tag{62}$$

It then follows that

$$f_{W_i}(0^+) = \lim_{w \to 0^+} \lambda(n-i)e^{-\lambda(n-i)w} = \lambda(n-i), \ \forall i, \tag{63}$$

$$f_{W_i, W_j}(\mathbf{0}_2^+) = f_{W_i}(0^+)f_{W_j}(0^+) = \lambda^2(n-i)(n-j), \ \forall i \neq j. \tag{64}$$

It is also easy to evaluate

$$f'_{W_i}(0^+) = \lim_{w \to 0^+} -\lambda^2(n-i)^2 e^{-\lambda(n-i)w} = -\lambda^2(n-i)^2, \ \forall i. \tag{65}$$

By substituting (63) and (65) into $c_1$ and $c_2$ in Corollay (3.7) with some algebras, we conclude the proof of Example 3.9.

## F  Proof of Corollary 3.11

Since $P_e(\phi_{\text{lin}}, \mathcal{K}) \to 0$ as $\sigma \to 0^+$, the first order rate is given from Theorem 3.6 by

$$P_e(\phi_{\text{lin}}, \mathcal{K}) = P_e^{(1)}\sigma + O(\sigma^2), \tag{66}$$

where

$$
\begin{aligned}
P_e^{(1)} &= \alpha_1(0^+) \\
&= \lim_{\sigma \to 0^+} \sum_{\substack{\mathcal{I} \subseteq [1:n-1] \\ |\mathcal{I}|=1}} \int_{\mathbf{u} \in \mathbb{R}_+^k} F_{\mathbf{V}_\mathcal{I}}(-\mathbf{u}) f_{\mathbf{W}_\mathcal{I}}(\sigma \mathbf{u}) \ \mathrm{d}\mathbf{u} \\
&= \lim_{\sigma \to 0^+} \sum_{i=1}^{n-1} \int_0^\infty \Pr(V_i \leq -u) f_{W_i}(\sigma u) \ \mathrm{d}u \\
&\overset{(a)}{=} \lim_{\sigma \to 0^+} \sum_{i=1}^{n-1} \int_0^\infty \Pr(V_1 \geq u) f_{W_i}(\sigma u) \ \mathrm{d}u \\
&\overset{(b)}{=} \sum_{i=1}^{n-1} f_{W_i}(0^+) \int_0^\infty \Pr(V_1 \geq u) \ \mathrm{d}u \\
&= \sum_{i=1}^{n-1} f_{W_i}\left(0^+\right) \int_0^\infty \mathbb{E}\left[\mathbb{1}_{\{V_1 > u\}}\right] \ \mathrm{d}u \\
&\overset{(c)}{=} \sum_{i=1}^{n-1} f_{W_i}\left(0^+\right) \mathbb{E}\left[\int_0^\infty \mathbb{1}_{\{V_1 > u\}} \ \mathrm{d}u\right] \\
&= \sum_{i=1}^{n-1} f_{W_i}\left(0^+\right) \mathbb{E}\left[\int_0^{\max\{V_1, 0\}} 1 \ \mathrm{d}u\right] \\
&= \sum_{i=1}^{n-1} f_{W_i}\left(0^+\right) \mathbb{E}\left[\max\{V_1, 0\}\right] \\
&\overset{(d)}{=} \frac{1}{2} \sum_{i=1}^{n-1} f_{W_i}(0^+) \mathbb{E}\left[|V_1|\right],
\end{aligned}
\tag{67}
$$

where the labeled equalities follow from: (a) the exchangeability of $\mathbf{N}$ (i.e., $V_i = N_{i+1} - N_i \overset{d}{=} N_i - N_{i+1} = -V_i$ and $V_i \overset{d}{=} V_j$, $\forall (i, j)$; (b) the dominated convergence theorem; (c) the Fubini-Tonelli theorem; and (d) the symmetry of $V$ as we have used in step (a). We conclude the proof of Corollary 3.11 by substituting (67) into (66).

## G    Proof of Proposition 3.14

For i.i.d. $X_i \sim F_X$, where $F_X$ is the CDF of $X$, we observe that Pyke (1965)

$$
\begin{aligned}
\sum_{i=1}^{n-1} f_{W_i}(0^+) &= \sum_{i=1}^{n-1} \frac{n!}{(i-1)!(n-i-1)!} \int_{-\infty}^{\infty} (F_X(x))^{i-1}(1 - F_X(x))^{n-i-1} f_X^2(x) \, dx \\
&\overset{(a)}{=} \int_{-\infty}^{\infty} \sum_{i=1}^{n-1} \frac{n!}{(i-1)!(n-i-1)!} (F_X(x))^{i-1}(1 - F_X(x))^{n-i-1} f_X^2(x) \, dx \\
&= \int_{-\infty}^{\infty} \sum_{i=1}^{n-1} \binom{n}{i-1} (n-i+1)(n-i)(F_X(x))^{i-1}(1 - F_X(x))^{n-i-1} f_X^2(x) \, dx \\
&\overset{(b)}{=} \int_{-\infty}^{\infty} \sum_{j=0}^{n-2} \binom{n}{j} (n-j)(n-j-1)(F_X(x))^j (1 - F_X(x))^{n-j-2} f_X^2(x) \, dx, \quad (68)
\end{aligned}
$$

where (a) follows by using the Fubini-Tonelli theorem, and (b) follows from the change of variable $j = i - 1$. To simplify the integrand in (68), we make use of the following,

$$
\begin{aligned}
&\sum_{j=0}^{n-2} \binom{n}{j} (n-j)(n-j-1)(F_X(x))^j (1 - F_X(x))^{n-j-2} \\
&= \sum_{j=0}^{n} \binom{n}{j} (n-j)(n-j-1)(F_X(x))^j (1 - F_X(x))^{n-j-2} \\
&\overset{(c)}{=} \mathbb{E}\left[(n-B)(n-B-1)\right](1 - F_X(x))^{-2} \\
&= \mathbb{E}\left[n^2 - 2nB + B^2 - n + B\right](1 - F_X(x))^{-2} \\
&= (n^2 - 2n^2 F_X(x) + nF_X(x)(1 - F_X(x)) + n^2 F_X^2(x) - n + nF_X(x))(1 - F_X(x))^{-2} \\
&= n(n - 2nF_X(x) + 2F_X(x) - F_X^2(x) + nF_X^2(x) - 1)(1 - F_X(x))^{-2} \\
&= n(n-1)(1 - 2F_X(x) + F_X^2(x))(1 - F_X(x))^{-2} \\
&= n(n-1), \quad (69)
\end{aligned}
$$

where in (c) we let $B \sim \text{Bin}(n, F_X(x))$ be the binomial random variable with parameters $n$ and $F_X(x)$, and the expectation is with respect to $B$.

Then, we have

$$
\begin{aligned}
\sum_{i=1}^{n-1} f_{W_i}(0^+) &= \int_{-\infty}^{\infty} n(n-1) f_X^2(x) dx \\
&= n(n-1) \int_{-\infty}^{\infty} f_X^2(x) dx.
\end{aligned}
$$

This concludes the proof of Proposition 3.14.

## H    Proof of Proposition 4.4

We consider the Laplace mechanism such that

$$
\mathcal{K}_L(\sigma, \mathbf{X}) = \mathbf{X} + \sigma \mathbf{N},
$$

where $\mathbf{N}$ is i.i.d. according to $\mathrm{Lap}(0, b)$. This result has already been shown by Mironov (2017) and we present it here for completeness. By using the definition of $\mathrm{RDP}_\alpha(\mathcal{K}_L)$ in Definition 2.2, we obtain

$$
\begin{aligned}
\mathrm{RDP}_\alpha(\mathcal{K}_L) &= \sup_{(\mathbf{X}, \tilde{\mathbf{X}}) \in \mathcal{X}^2 : d_H(\mathbf{X}, \tilde{\mathbf{X}}) \leq 1} D_\alpha(\mathcal{K}(\mathbf{X}) \| \mathcal{K}(\tilde{\mathbf{X}})) \\
&\overset{(a)}{=} \sup_{|x_1 - x_2| \leq \ell} D_\alpha(\mathrm{Lap}(x_1, \sigma b) \| \mathrm{Lap}(x_2, \sigma b)) \\
&= \sup_{r \in [0, \ell]} D_\alpha(\mathrm{Lap}(0, \sigma b) \| \mathrm{Lap}(r, \sigma b)) \\
&\overset{(b)}{=} \sup_{r \in [0, \ell]} \frac{1}{\alpha - 1} \ln \frac{\alpha e^{-(1-\alpha)r/(\sigma b)} - (1-\alpha)e^{-\alpha r/(\sigma b)}}{2\alpha - 1} \\
&\overset{(c)}{=} \frac{1}{\alpha - 1} \ln \frac{\alpha e^{-(1-\alpha)\ell/(\sigma b)} - (1-\alpha)e^{-\alpha \ell/(\sigma b)}}{2\alpha - 1},
\end{aligned}
\tag{70}
$$

where the labeled equalities follow from: (a) the fact that $\mathcal{K}(\mathbf{X})$ and $\mathcal{K}(\tilde{\mathbf{X}})$ have nearly identical distributions that differ at only one coordinate; (b) the closed-form expression by Gil et al. (2013); and (c) the fact that $D_\alpha(\mathrm{Lap}(0, \sigma b) \| \mathrm{Lap}(r, \sigma b))$ is an increasing function in $r$. Therefore, $\mathcal{K}_L(\sigma, \mathbf{X})$ gives $(\alpha, \epsilon)$-RDP with $\epsilon$ given in (70).

For the upper and lower bounds, we first obtain an upper bound by using the concavity property of the logarithm and its first-order condition (Boyd et al., 2004), i.e.,

$$
\ln(x + y) \leq \ln(x) + \frac{y}{x}, \quad \forall y, x > 0.
\tag{71}
$$

Using the above inequality, we can upper bound (70) as

$$
\begin{aligned}
\epsilon &= \frac{1}{\alpha - 1} \ln \frac{\alpha e^{-(1-\alpha)\ell/(\sigma b)} - (1-\alpha)e^{-\alpha \ell/(\sigma b)}}{2\alpha - 1} \\
&\leq \frac{1}{\alpha - 1} \left( \ln \frac{\alpha e^{(\alpha-1)\ell/(\sigma b)}}{2\alpha - 1} + \frac{(\alpha - 1)e^{-\alpha \ell/(\sigma b)}}{\alpha e^{(\alpha-1)\ell/(\sigma b)}} \right) \\
&= \frac{1}{\alpha - 1} \ln \frac{\alpha e^{(\alpha-1)\ell/(\sigma b)}}{2\alpha - 1} + \frac{1}{\alpha} e^{-\frac{(2\alpha-1)\ell}{\sigma b}} \\
&= \frac{\ell}{\sigma b} + \frac{1}{\alpha - 1} \ln \frac{\alpha}{2\alpha - 1} + \frac{1}{\alpha} e^{-\frac{(2\alpha-1)\ell}{\sigma b}}.
\end{aligned}
\tag{72}
$$

A lower bound can be obtained by dropping the second exponential term in (70) as

$$
\begin{aligned}
\epsilon &\geq \frac{1}{\alpha - 1} \ln \frac{\alpha e^{-(1-\alpha)\ell/(\sigma b)}}{2\alpha - 1} \\
&= \frac{\ell}{\sigma b} + \frac{1}{\alpha - 1} \ln \frac{\alpha}{2\alpha - 1}.
\end{aligned}
\tag{73}
$$

This concludes the proof of Proposition 4.4.

## I  Proof of Corollary 4.5

From the lower bound in (28) in Proposition 4.4, we directly obtain the lower bound of $\sigma$ as

$$
\frac{\ell}{b \left( \epsilon + \frac{1}{\alpha - 1} \ln \frac{2\alpha - 1}{\alpha} \right)} \leq \sigma.
\tag{74}
$$

In addition, the upper bound in (28) can be further bounded by

$$
\epsilon \leq \frac{\ell}{\sigma b},
\tag{75}
$$

which follows from the fact that $\frac{1}{\alpha-1} \ln \frac{\alpha}{2\alpha-1} + \frac{1}{\alpha} e^{-\frac{(2\alpha-1)\ell}{\sigma b}}$ is increasing in $\alpha > 1$ for any values of $\sigma > 0, b > 0$, and $\ell \geq 0$, and the limit is 0, i.e.,

$$\lim_{\alpha \to \infty} \frac{1}{\alpha-1} \ln \frac{\alpha}{2\alpha-1} + \frac{1}{\alpha} e^{-\frac{(2\alpha-1)\ell}{\sigma b}} = 0.$$

The bound in (75) gives the upper bound on $\sigma$ as

$$\sigma \leq \frac{\ell}{b\epsilon}, \tag{76}$$

and hence, with (74) we have that, for the Laplace mechanism,

$$\frac{\ell}{b\left(\epsilon + \frac{1}{\alpha-1} \ln \frac{2\alpha-1}{\alpha}\right)} \leq \mathsf{RDP}_\alpha^{-1}(\epsilon, \ell) \leq \frac{\ell}{b\epsilon}. \tag{77}$$

We now leverage Proposition 4.2, which requires $\mathbb{E}[|V|]$. In order to have $\mathrm{Var}(N) = 1$ for $N \sim \mathrm{Lap}(0, b)$, we set $b = \frac{1}{\sqrt{2}}$, and we evaluate $\mathbb{E}[|V|]$ as follows. The PDF of $V = N_1 - N_2$, where $N_1$ and $N_2$ are independent $\mathrm{Lap}\left(0, \frac{1}{\sqrt{2}}\right)$, is given by

$$\begin{aligned} f_V(v) &= \int_{-\infty}^{\infty} f_{N_1}(z) f_{N_2}(v-z) \, dz \\ &= \int_{-\infty}^{\infty} \frac{1}{\sqrt{2}} e^{-\sqrt{2}|z|} \frac{1}{\sqrt{2}} e^{-\sqrt{2}|v-z|} \, dz \\ &= \frac{1}{2\sqrt{2}} e^{-\sqrt{2}|v|} + \frac{1}{2}|v| e^{-\sqrt{2}|v|}. \end{aligned} \tag{78}$$

Then, by the symmetry of $V$ we have

$$\mathbb{E}[|V|] = 2 \int_0^\infty v f_V(v) \, dv = \frac{3}{2\sqrt{2}}. \tag{79}$$

We obtain the trade-off expression by combining (77) and (25) as

$$P_e(\phi_{\mathrm{lin}}, \mathcal{K}) = \frac{3C_{\mathbf{X}}}{4\sqrt{2}} \mathsf{RDP}_\alpha^{-1}(\epsilon, \ell) + O\left(\frac{1}{\epsilon^2}\right), \tag{80}$$

where

$$\frac{\sqrt{2}\ell}{\epsilon + \frac{1}{\alpha-1} \ln \frac{2\alpha-1}{\alpha}} \leq \mathsf{RDP}_\alpha^{-1}(\epsilon, \ell) \leq \frac{\sqrt{2}\ell}{\epsilon}. \tag{81}$$

This concludes the proof of Corollary 4.5.

## J   Proof of Proposition 4.8

Consider the Gaussian mechanism such that

$$\mathcal{K}_G(\sigma, \mathbf{X}) = \mathbf{X} + \sigma \mathbf{N},$$

where $\mathbf{N}$ is i.i.d. according to $\mathcal{N}(0,1)$. By using the definition of $\mathrm{RDP}_\alpha(\mathcal{K}_G)$ in Definition 2.2, we obtain

$$
\begin{aligned}
\mathrm{RDP}_\alpha(\mathcal{K}_G) &= \sup_{(\mathbf{X},\tilde{\mathbf{X}})\in\mathcal{X}^2:d_H(\mathbf{X},\tilde{\mathbf{X}})\leq 1} D_\alpha(\mathcal{K}(\mathbf{X})\|\mathcal{K}(\tilde{\mathbf{X}})) \\
&\overset{(a)}{=} \sup_{|x_1-x_2|\leq\ell} D_\alpha(\mathcal{N}(x_1,\sigma^2)\|\mathcal{N}(x_2,\sigma^2)) \\
&= \sup_{r\in[0,\ell]} D_\alpha(\mathcal{N}(0,\sigma^2)\|\mathcal{N}(r,\sigma^2)) \\
&\overset{(b)}{=} \sup_{r\in[0,\ell]} \frac{1}{2}\frac{\alpha r^2}{\sigma^2} \\
&\overset{(c)}{=} \frac{1}{2}\frac{\alpha\ell^2}{\sigma^2},
\end{aligned}
\tag{82}
$$

where the labeled equalities follow from: (a) the fact that $\mathcal{K}(\mathbf{X})$ and $\mathcal{K}(\tilde{\mathbf{X}})$ have nearly identical distributions that differ at only one coordinate; (b) the closed-form expression by Gil et al. (2013); and (c) the fact that $\frac{\alpha r^2}{\sigma^2}$ is an increasing function in $r$. Using (24), we obtain

$$
\mathrm{RDP}_\alpha^{-1}(\epsilon,\ell) = \sqrt{\frac{\alpha\ell^2}{2\epsilon}}.
\tag{83}
$$

This concludes the proof of Proposition 4.8.

## K   Proof of Proposition 4.11

We start by noting that $\sigma N$ with $N \sim \mathcal{GN}(0,h(p),p)$ has variance equal to $\sigma^2$ and PDF given by Nadarajah (2005),

$$
f_{\sigma N}(z) = K\exp\left(-\left|\frac{z}{\sigma h(p)}\right|^p\right),
\tag{84}
$$

where $h(p) = \sqrt{\frac{\Gamma(p^{-1})}{\Gamma(3p^{-1})}}$ and $K = \frac{p}{2\sigma h(p)\Gamma(p^{-1})}$.

Since $(\infty,\epsilon)$-RDP is equivalent to $\epsilon$-DP, we evaluate the Rènyi divergence of order $\alpha=\infty$. From Definition 2.2, the Rènyi divergence of order $\alpha=\infty$ between $\mathbf{x}+\sigma\mathbf{N}$ and $\tilde{\mathbf{x}}+\sigma\mathbf{N}$ with $d_H(\mathbf{x},\tilde{\mathbf{x}})\leq 1$ is

$$
\begin{aligned}
D_\infty(\mathcal{K}(\mathbf{x}+\sigma\mathbf{N})\|\mathcal{K}(\tilde{\mathbf{x}}+\sigma\mathbf{N})) &\overset{(a)}{=} D_\infty(\mathcal{K}(r+\sigma N)\|\mathcal{K}(\sigma N)) \\
&\overset{(b)}{=} \sup_{z\in\mathbb{R}} \log\frac{f_{\sigma N}(z-r)}{f_{\sigma N}(z)} \\
&= \sup_{z\in\mathbb{R}}\left\{-\left|\frac{z-r}{\sigma h(p)}\right|^p + \left|\frac{z}{\sigma h(p)}\right|^p\right\} \\
&\overset{(c)}{=} \left|\frac{r}{\sigma h(p)}\right|^p,
\end{aligned}
\tag{85}
$$

where the labeled equalities follow from: (a) the fact that $r\in[-\ell,\ell]$ is the difference between $\mathbf{x}$ and $\tilde{\mathbf{x}}$, and without loss of generality we consider positive $r\in[0,\ell]$ due to the symmetry of $N$; (b) the definition of the Rènyi divergence of order $\alpha=\infty$; and (c) the fact that the maximum of the function $t\mapsto|t|^p-|t-r|^p$ is obtained at $t=r$ for $0<p<1$.

Since (85) is an increasing function in $r\in[0,\ell]$, we obtain

$$
\mathrm{RDP}_\infty(\mathcal{K}_{GN}) = \sup_{r\in[0,\ell]}\left|\frac{r}{\sigma h(p)}\right|^p = \frac{1}{\sigma^p}\left(\frac{\ell}{h(p)}\right)^p.
\tag{86}
$$

This concludes the proof of Proposition 4.11.

## L   Minimizing $(35)$ **with respect to** $0 < p \leq 1$

To find the minimum value (or minimizer) of $\frac{1}{h(p)} \left(\frac{1}{\epsilon}\right)^{\frac{1}{p}}$ with respect to $0 < p \leq 1$, we differentiate it with respect to $p$ and obtain

$$
\frac{\partial}{\partial p} \left\{ \frac{1}{h(p)} \left(\frac{1}{\epsilon}\right)^{\frac{1}{p}} \right\} = \frac{\partial}{\partial p} \left\{ \sqrt{\frac{\Gamma(3p^{-1})}{\Gamma(p^{-1})}} \left(\frac{1}{\epsilon}\right)^{\frac{1}{p}} \right\}
$$

$$
= \frac{\partial p^{-1}}{\partial p} \frac{\partial}{\partial p^{-1}} \left\{ \sqrt{\frac{\Gamma(3p^{-1})}{\Gamma(p^{-1})}} \left(\frac{1}{\epsilon}\right)^{\frac{1}{p}} \right\}
$$

$$
\stackrel{(a)}{=} \frac{\partial p^{-1}}{\partial p} \frac{\partial}{\partial x} \left\{ \sqrt{e^{\ln \Gamma(3x) - \ln \Gamma(x)}} \left(\frac{1}{\epsilon}\right)^{x} \right\}
$$

$$
\stackrel{(b)}{=} -\frac{1}{p^2} \frac{(3\psi(3x) - \psi(x) - 2\ln \epsilon)}{2} \sqrt{\frac{\Gamma(3x)}{\Gamma(x)}} \left(\frac{1}{\epsilon}\right)^{x}
$$

$$
= -\frac{1}{2p^2} \sqrt{\frac{\Gamma(\frac{3}{p})}{\Gamma(\frac{1}{p})}} \left(\frac{1}{\epsilon}\right)^{\frac{1}{p}} \left(3\psi(3p^{-1}) - \psi(p^{-1}) - 2\ln \epsilon\right), \tag{87}
$$

where the labeled equalities follow from: (a) the change of variable $x = p^{-1}$; and (b) letting $\psi(x) = \frac{\mathrm{d}}{\mathrm{d}x} \ln \Gamma(x) = \frac{\Gamma'(x)}{\Gamma(x)}$ be the digamma function (Abramowitz & Stegun, 1964, p.258).

By using the change of variable $x = p^{-1} \in [1, \infty)$, we have an equivalent expression for the derivative,

$$
-\frac{x^2}{2} \sqrt{\frac{\Gamma(3x)}{\Gamma(x)}} \left(\frac{1}{\epsilon}\right)^{x} \left(3\psi(3x) - \psi(x) - 2\ln \epsilon\right). \tag{88}
$$

Since the sign of $-\frac{x^2}{2} \sqrt{\frac{\Gamma(3x)}{\Gamma(x)}} \left(\frac{1}{\epsilon}\right)^{x}$ is negative for all $x \in [1, \infty)$, it is sufficient to consider the last term $(3\psi(3x) - \psi(x) - 2\ln \epsilon)$. The digamma function $\psi(x)$ does not have a closed-form expression and hence, we instead use the approximation $\psi(x) \approx \ln x - \frac{1}{cx}$, where $1 \leq c \leq 2$ is a constant. This approximation expression comes from the following bounds (Minc & Sathre, 1964, Lemma 2) for $x \geq 1$,

$$
\ln x - \frac{1}{x} \leq \psi(x) \leq \ln x - \frac{1}{2x}. \tag{89}
$$

Using $\psi(x) \approx \ln x - \frac{1}{cx}$, we have that

$$
3\psi(3x) - \psi(x) - 2\ln \epsilon \approx \ln(27x^2) - \ln \epsilon^2, \tag{90}
$$

which is increasing in $x$. Hence, $(88)$ is negative if $\frac{\epsilon}{3\sqrt{3}} < x$ and is positive otherwise, which implies that the minimum value of $\frac{1}{h(p)} \left(\frac{1}{\epsilon}\right)^{\frac{1}{p}}$ can be obtained by choosing $p$ such that $\frac{1}{p} = x \approx \frac{\epsilon}{3\sqrt{3}}$. Due to the condition $p \leq 1$, we finally obtain the approximed minimizer $p$ for $\frac{1}{h(p)} \left(\frac{1}{\epsilon}\right)^{\frac{1}{p}}$ given by

$$
\hat{p} = \min \left\{ \frac{3\sqrt{3}}{\epsilon}, 1 \right\}. \tag{91}
$$

Note that an exact expression for the minimizer is $p = \min\{p', 1\}$ where $p'$ is such that

$$
3\psi \left(\frac{3}{p'}\right) - \psi \left(\frac{1}{p'}\right) - 2\ln \epsilon = 0, \tag{92}
$$

which can be obtained numerically.

