# OpenReview forum: "Ranking Recovery under Privacy Considerations"
_TMLR — Accepted by TMLR_

### Review · Reviewer_NH5D · 2022-06-01

**Summary Of Contributions:**

The paper considers "the private ranking recovery problem, where a data collector seeks to estimate the permutation/ranking of a data vector given a randomized (privatized) version of it."

It considers the Rényi differential privacy (RDP) for a mechanism that for input data vector X adds noise N, where N is $\ell_p$ - spherical noise, in particular '$\ell_p$ - spherically non-increasing'. This is related to the 'ranking recovery' problem presented in

Minoh Jeong, Alex Dytso, and Martina Cardone. Retrieving data permutations from noisy observations: High and low noise asymptotics. In Proceedings of the 2021 IEEE International Symposium on Information Theory (ISIT), July 2021,

upon which this works seems to build (lot of similar notation used as well).

Approximations for the $\varepsilon$-RDP values of various noise adding mechanisms are given, as well as few strict bounds.

**Broader Impact Concerns:**

This is theory paper and would potentially improve differential privacy techniques, I do not see anything reason for having a Broader Impact Statement.

**Requested Changes:**

I think the paper would need a major revision before being ready for publication.

First of all, the results should be rewritten: I do not see a reason for having the error function in the RDP bounds if the noise adding mechanism is the one that is being analysed. Also, here, the property of post-processing of DP (and of RDP, due to data processing inequality) would be useful. Otherwise, I would imagine that the error function $P_e$ could be useful in case the output of the RDP mechanism would be the ranking, for example.

The $\varepsilon$-approximations of Section 4 should be replaced by upper bounds. In that case the expansion of $P_e$ given in Thm 3.6 might be less useful as it seems to be quite a complicated expression.

There should be some illustration of the tightness of the obtained bounds.

All of these are crucial in my opinion, the paper is not ready for publication.

**Strengths And Weaknesses:**

Strengths:

- The approach based on the analysis of the error estimation probability $P_e$ (Eq. 3) for rankings of noisy vectors seems interesting and might be useful addition to DP literature.

- The paper is mostly well written, easy to follow.

Weaknesses:

- I think the main weakness is that the contribution seems quite thin. The only valid RDP bounds are those of Prop. 4.1 (simply a Laplace mechanisms RDP, given already in reference Mironov, 2017), and that of Prop. 4.7. The bound of Prop. 4.7 is not really an RDP bound as it only gives the cases $\alpha=1$ and $\alpha=\infty$, i.e. the KL and pure eps-DP bound.

- Some of the results are hard to read. I cannot understand why the error function $P_e$ appears in the $\varepsilon$-approximations of Section 4. As far as I see, looking at eq. 2, the noise adding mechanism which is analysed in these results is inpendent of the error function $P_e$. It is quite uncommon in DP literature to give approximations for epsilon-values (security mindset, we should always report upper bounds), and especially here I think it is very unclear how rough these approximations are.

- The basics of DP and RDP should be introduced more carefully. I think that the claim on. p 2 that RDP encompasses eps-DP and (eps,delta)-DP is wrong.

---

> ### Author Response · Authors · 2022-06-19
> **Response to Reviewer NH5D**
>
> We thank the Reviewer for his/her careful reading of our manuscript and his/her efforts in reviewing the paper. We are glad that the writing of the paper has been to the Reviewer’s satisfaction. We hope that our answer to the points below properly addresses the Reviewer’s concerns and comments.
>
> 1) Regarding the fact that the contribution seems quite thin.
> The Reviewer is pointing out that the results in Section 4 of our paper do not provide any novel bounds on the RDP or $\epsilon$-DP. Indeed, this is the case, and there are only a few new bounds on  RDP or $\epsilon$-DP that are not difficult to obtain.    However, our goal was not to provide any new bounds on the RDP or $\epsilon$-DP, but to evaluate Proposition 4.2 for several important commonly used mechanisms.   We have added a paragraph at the end of  Section 4.1 to make this point clear.  In our work, we view RDP or $\epsilon$-DP, as already well-understood and commonly used measures of privacy. Therefore, in this paper, we use these as standard measures to understand the privacy leakage in the ranking recovery problem. Our main contribution is to explain how an imposed privacy constraint in terms of R\'enyi DP or $\epsilon$-DP interacts with other aspects of the problem, such as the prior distribution of the data, the dimensionality of the data, and the nature of the additive noise. For example, we have shown a clear trade-off between the size of the data and privacy. Specifically, we have shown that there can be a quadratic dependence on the data size, which gives a clear indication on how the privacy measure $\epsilon$ needs to be adjusted if we fix the utility but increase/decrease the data size. Our results have also been used to argue what noise mechanisms are the best for this setting. In particular, we were able to argue that for a given privacy level, heavy tail distributions provide a better privacy-utility trade-off than, for example, a Laplace mechanism. This observation is interesting because the Laplace mechanism was shown to be nearly optimal (more specifically, the staircase mechanism is optimal) for linear utility functions.
> Finally, we would like to highlight that all these results would not have been possible without our rigorous analysis on the probability of error (Section 3), which we consider the bulk of our contribution.
>
> 2) Regarding $P_e$ in the $\epsilon$ expressions. We start by noting that both the probability of error (i.e., utility) expression in Corollary 3.11 and the RDP in Definition 2.2 are a function of the noise standard deviation $\sigma$. This implies that there is indeed a relationship between $P_e$ and RDP. We formalized such a relationship by using the definition in eq.(24), which provides the smallest standard deviation of the randomized mechanism that ensures to meet the $(\alpha,\epsilon)$-RDP. Then, the probability of error expression in eq.(25) inside Proposition 4.2 is evaluated using such a smallest standard deviation. It, therefore, follows that eq.(25) ties together the probability of error and $\epsilon$ (since eq.(24) is a function of $\epsilon$), and this gives rise to expressions of $\epsilon$ that indeed depend on the probability of error. Such expressions represent the privacy-utility trade-offs. We hope that this discussion properly clarifies the concern of the Reviewer. Regarding this point, the Reviewer also points out that the property of post-processing of DP (and of RDP, due to data processing inequality) would be useful in the analysis. However, we could not understand this point (i.e., how can data-processing inequality be helpful in the analysis?), and we would appreciate it if the Reviewer could please expand on this.
>
> 3) Regarding the approximations on the $\epsilon$ values. We thank the Reviewer for this comment. In the new version of the manuscript, we have revised  Corollary 4.5, Corollary 4.9, and Corollary 4.13, which now provide exact (and not anymore approximate) expressions for the value of $\epsilon$  for the Laplace mechanism, the Gaussian mechanism and the generalized normal mechanism, respectively. Moreover, we also point out that, while Corollary 4.9 (Gaussian mechanism) and  Corollary 4.13 (generalized normal mechanism), provide the privacy-utility trade-off in a simple form (i.e., they show that $\epsilon$ decreases as $1/P_e^p$), the result in Corollary 4.5 (Laplace mechanism) might not be easy to interpret. Because of this, we have included eq.(28) in Proposition 4.4 that provides a lower bound and an upper bound on $\epsilon$. These bounds show that for the Laplace mechanism $\epsilon$ decreases as $1/P_e$.
>
> 4) Regarding the basics of DP and RDP. We thank the Reviewer for pointing this out, and we agree that our claim in the original version of the manuscript was not correct. In the revised version, on page 2 we have written: ``Moreover, as pointed out in (Mironov, 2017, Proposition 3), the $(\alpha,\epsilon)$-RDP can be converted to the $(\epsilon,\delta)$-DP.''

---

### Review · Reviewer_P18b · 2022-06-03

**Summary Of Contributions:**

The paper studied private ranking recovery problem by using Gaussian, Laplace and generalized normal model as randomizer under Renyi differential privacy. The authors aim to establish the trade-off between privacy and utility by measuring the probability error in estimation and the level of privacy in RDP. When linear decoder is employed, they derive Taylor series of the probability of error, and use first and second-order approximations to characterize trade-offs between privacy and probability of error.



**Requested Changes:**

The authors should give some experiments on real world data to verify the theoretical results.



**Strengths And Weaknesses:**

Strengths:
I think the problem of the paper is interesting and important. The authors nicely combine ranking problem and the differential privacy tools to protect the data privacy. The authors have adequately cited related work as well as honestly commented on differences and similarities with existing literature. The analysis is not entirely novel but it is definitely interesting.
The submission looks technically sound and makes a solid theoretical contribution. And the authors kindly give some examples to make the results more readable.  I did go through the proofs in detail and they seem to be correct. It is good to bring some simulations results for different distributions of input data.

Weaknesses:
The authors give three upper bounds for trade-off, but there is no evidence (such as lower bound) to show whether they are optimal. The authors should give some experiments on real world data to verify the theoretical results.

---

> ### Author Response · Authors · 2022-06-19
> **Response to Reviewer P18b**
>
> We thank the Reviewer for his/her thorough reading of our manuscript and appreciation of our analysis and contribution.
> We hope that our answer to the points below properly addresses the Reviewer’s concerns and comments.
>
> 1) Regarding the upper bounds on the trade-off. We thank the Reviewer for this comment. In the new version of the manuscript, we have revised and strengthened our privacy-utility trade-off results, and showed their tightness. In particular, Corollary 4.5, Corollary 4.9, and Corollary 4.13 present the trade-off between privacy and utility for the Laplace mechanism,  Gaussian mechanism, and generalized normal mechanism, respectively, with the appropriate error and bounds clearly stated.}  Moreover, we also point out that, while Corollary 4.9 (Gaussian mechanism) and Corollary 4.13 (generalized normal mechanism), provide the privacy-utility trade-off in a simple form (i.e., they show that $\epsilon$ decreases as $1/P_e^p$), the result for the Laplace does not have a closed-form expression for the inverse in eq.(24) and might not be easy to interpret. {Because of this, we have included eq.(28) in Proposition 4.4 which provides a lower bound and an upper bound on $\epsilon$.
>
>
> 2) Regarding experiments on real-world data to verify the theoretical results. We agree with the Reviewer that such experiments would add value to the paper. However, we leave these as future work. We indeed agree with the Editor that such experiments can not be deemed as essentials for a theory-focused paper. We hope that our choice (also highlighted by the Editor) to prioritize the derivation of the new results that show the tightness of the trade-off between privacy and utility (i.e., Corollary 4.5, Corollary 4.9, and Corollary 4.13) will be appreciated by the Reviewer.

---

### Review · Reviewer_iKza · 2022-06-07

**Summary Of Contributions:**

The submission studies the problem of recovering the ranking of data vector entries given a privatized version of the data. The main contributions are the following: 1) showing that a simple, computationally efficient mechanism (in a way, the “obvious” one) for ranking estimation is optimal under potentially restrictive but reasonable assumptions; 2) approximating the probability of error in ranking recovery via a Taylor series argument; 3) (heuristically) quantifying the tradeoff between Renyi differential privacy and probability of error given a Taylor approximation.

**Broader Impact Concerns:**

I don’t think the paper raises broader impact concerns.

**Requested Changes:**

The requested changes are outlined in the previous section. Point 2 is the most critical to address.

**Strengths And Weaknesses:**

The problem setting is well motivated and the presentation is generally clear. I checked the proof of Theorem 3.3 and it makes sense and is correct to the best of my knowledge. The weaknesses I see are the following:
1. The privacy part is really only relevant at the end, and moreover the privacy-utility tradeoff is a bit heuristic (as I will elaborate), so the paper can be thought of as just dealing with ranking recovery from a noisy data vector (where the noise has to satisfy certain regularity properties). In this way the problem setting seems to be the same as what was considered by Jeong et al. Remark 3.4 clarifies that there are some technical improvements over these prior results, so it seems that the paper checks the novelty box.
2. Many technical statements seem imprecise and too informal for my taste. For example, at the beginning of the proof of Theorem 3.3, it says that “the maximum likelihood decoder is optimal [for minimizing the probability of incorrect recovery]” and there is a reference to a whole book. While I don’t think there is a bug in the proof, this is a kind of statement that requires either a more precise reference or a proof. Please formalize. The results in Section 4 are likewise informal; since the probability of error is only approximated using a Taylor series argument, I wouldn’t call statements such as Theorem 4.2 and Theorem 4.5 “theorems”. Please either clarify that this comparison is heuristic given that an approximation for the error is used, or make a rigorous mathematical statement.
3. While the paper positions itself as a privacy paper, the actual privacy discussion is fairly trivial. In particular, the non-trivial analysis is in analyzing ranking recovery from noisy data (which is not a newly analyzed problem). At the end the privacy only comes in when relating the noise level to a privacy parameter, and these identities are well known. Please cite the corresponding results in Mironov (2017).
4. There are several small points that need or are worth addressing:

a) For example, in Section 4 sensitivity is defined as l1-sensitivity but the different noise mechanisms require different sensitivity measures (Gaussian requires l2—I know l1 bounds l2 but relying on this bound seems a bit sloppy).

b) There is no sigma anywhere in the proof of Theorem 3.3 even though it’s a parameter of the randomization, please fix.

c) It is remarked that the probability of error grows quadratically in n; what is preventing this value from exceeding 1? For example, in Example 3.8, as we let n tend to infinity wouldn’t c1 go to infinity and thus Pe>1?

d) In the definition of alpha_k(sigma) on page 7, I was confused by the use of sigma as the argument since you’ve been using sigma as the noise magnitude.

e) The correct reference for DP is “Calibrating noise to sensitivity in private data analysis”, Dwork, McSherry, Nissim, Smith (2006) (top of page 2 in the submission).

5. I don’t understand the motivation for studying the generalized normal model. Can you give some important references in DP that study this noise model? It doesn’t seem nearly as fundamental as the other two and the analysis for it is looser.
6. I don’t understand the comment saying “private ranking recovery is noise dominated”; please elaborate.

---

> ### Author Response · Authors · 2022-06-19
> **Response to Reviewer iKza (1/2)**
>
> We thank the Reviewer for his/her careful reading of the manuscript and his/her efforts in reviewing the
> paper. We are glad that the problem setting and the writing of the paper have been to the Reviewer's satisfaction. We are also happy that the Reviewer appreciated Remark 3.4 which highlights the key novelties (both in the result and in the proof) of Theorem 3.3 with respect to the work by Jeong et al.
>
> We hope that our answer to the points below properly addresses the Reviewer’s concerns and comments.
>
> 1) Regarding the optimality of the maximum likelihood criterion. In the revised version of the manuscript on page 6, we have included the formal proof of this statement.
>
> 2) Regarding the presentation of the results in Section 4. We agree with the Reviewer that Theorem 4.2 and Theorem 4.5 in the original version of the manuscript (Corollary 4.5 and Corollary 4.9 in the revised version of the paper) were not very rigorous.  {In the revised version of the manuscript, Corollary 4.5 and Corollary 4.9 present the trade-off between privacy and utility for the Laplace mechanism and for the Gaussian mechanism, respectively, with the appropriate error and bounds clearly stated.  In particular, to obtain these results, we have taken into consideration the approximation error (resulting from the Taylor approximation) inside eq.(25), where we stated the error in terms of big-O notation. Furthermore, for the Laplace mechanism, in Proposition 4.4. and Corollary 4.5, we have provided rigorous bounds on the utility-privacy trade-off and not approximation (i.e., $\approx)$ as was done in the original version of the paper.
>    In other words, Corollary 4.5 and Corollary 4.9 now present exact results and not heuristics as in the original version of the manuscript.
>
>
> 3) Regarding the triviality of the privacy discussion.
> We agree with the Reviewer that we have not provided any novel bounds or evaluations of the R\'enyi DP or $\epsilon$-DP.  In our work, we viewed these quantities as already well-understood and commonly used measures of privacy. Therefore, in our work, we used these as standard measures to understand the privacy leakage in the ranking recovery problem. Our main contribution was to explain how an imposed privacy constraint in terms of R\'enyi DP or $\epsilon$-DP interacts with other aspects of the problem, such as the prior distribution of the data, the dimensionality of the data, and the nature of the additive noise. For example, we have shown a clear trade-off between the size of the data and privacy. Specifically, we have shown that there can be a quadratic dependence on the data size, which gives a clear indication on how the privacy measure $\epsilon$ needs to be adjusted if we fix the utility but increase/decrease the data size. Our results have also been used to argue what noise mechanisms are the best for this setting. In particular, we were able to argue that for a given privacy level, heavy tail distributions provide a better privacy-utility trade-off than, for example, a Laplace mechanism. This observation is interesting because the Laplace mechanism was shown to be nearly optimal (more specifically, the staircase mechanism is optimal) for linear utility functions.
>  Finally, according to the Reviewer's suggestion, we have properly cited and commented on the results that were derived by Mironov (2017).
>
>
> 4) Regarding the $\ell_1$ sensitivity. We thank the Reviewer for pointing this out, and we indeed acknowledge that different noise mechanisms may require different sensitivity measures. In the revised version of the manuscript, we have considered the $\ell_p$ sensitivity (formally defined in Definition 4.1) where the parameter $p>0$. However, as also discussed after Definition 4.1, in our setting we have that the query is the identity function, and this implies that the $\ell_p$ sensitivity is the same for all values of $p>0$.
>
> 5) Regarding $\sigma$ inside the proof of Theorem 3.3. We thank the Reviewer for pointing this out. In the revised version of the manuscript, we have incorporated the parameter $\sigma$ whenever the randomized mechanism $\mathcal{K}$ is used (as defined in eq.(2)).

---

> ### Author Response · Authors · 2022-06-19
> **Response to Reviewer iKza (2/2)**
>
> 5) Regarding $\sigma$ inside the proof of Theorem 3.3. We thank the Reviewer for pointing this out. In the revised version of the manuscript, we have incorporated the parameter $\sigma$ whenever the randomized mechanism $\mathcal{K}$ is used (as defined in eq.(2)).
>
> 6) Regarding the probability of error exceeding one. We thank the Reviewer for pointing out this source of confusion. We note that the first and second-order rates of the probability of error in Corollary 3.7 were derived in the limit of $\sigma$ that tends to $0$ (see Appendix D). Thus, in Corollary 3.7, even though $c_1$ can grow unboundedly as $n$ grows, it is multiplied by $\sigma$ that is small. In the revised version of the manuscript, we have highlighted that the second-order approximation of the probability of error in Corollary 3.7 holds in the low-noise regime.
>
> 7) Regarding $\alpha_k(\sigma)$ in Theorem 3.6. We thank the Reviewer for pointing out this source of confusion. To avoid such confusion, in the revised version of the manuscript, we used the notation $\alpha_k(\omega)$.
>
> 8) Regarding the reference on DP. We thank the Reviewer for this comment. In the revised version of the manuscript, we have replaced the previous reference with the one pointed out by the Reviewer.
>
> 9) Regarding the motivation on studying the generalized Gaussian model. In the revised version of the manuscript, we have included the reference (Liu, 2018), where the author studied the generalized Gaussian mechanism for DP. We have also added Remark 4.12 in which we briefly discuss the contribution of (Liu, 2018) and we highlight the novelty of our contribution in Proposition 4.11 for the generalized normal mechanism.
>
> 10) Regarding the comment saying ''private ranking recovery is noise dominated''. On page 2 of the manuscript, we explained this by saying "This result suggests that the private ranking recovery problem is noise dominated, i.e., the error probability is large even for small values of the noise variance.'' In the revised version of the manuscript, we also added this explanation in Section 5 on page 14. We would be happy to rephrase this sentence, should the Reviewer have any suggestions on the language to use.

---

### Comment · Action_Editors · 2022-06-07
**Discussion Period**

Thanks to all reviewers to submitting their review. We are now in the post-review discussion period.

The goal of the reviewers should be to gather all the information they need to be comfortable submitting a decision recommendation for this submission. To remind, the possible decision recommendations are Accept, Accept with minor revision, and Reject (where reviewers are additionally given the option to mark that they would consider a significant revision -- essentially a "revise and resubmit" recommendation).

The reviewers seem to ask for changes in three different ways: iKza wants the statements to be more precise and formal, P18b wants empirical evaluation, and NH5D wants some additional results and rewriting. There are also some general comments about the contribution of the paper being slim or not the most novel. To interject my own opinions: while good and thorough experiments of course make any paper even better, this seems to be primarily a theory-focused paper, and I don't see them as essential. Reviewer P18b can speak up if they disagree on any of these points. I would find P18b's other suggestion of a lower bound (also hinted at by NH5D) to be more appropriate, if provable.

If the authors could please try their best to a) revise their manuscript according to the feedback given and b) update the authors about how they have addressed their comments, that would be great. Note that recommendations from the reviewers may be submitted in 2 weeks from now, and must be submitted at most 4 weeks from now.

If the reviewers have any further questions at this point, feel free to ask them to the authors. They can additionally discuss the merits of this paper privately among themselves. In deliberating the paper, please recall the Evaluation criteria as described here (https://jmlr.org/tmlr/editorial-policies.html).

---

> ### Author Response · Authors · 2022-06-19
> **Response to the Action Editor**
>
> We would like to thank the Editor and the Reviewers for their careful reading of our manuscript, their detailed
> comments, their suggestions on how to improve the presentation of the material, and their criticisms. In the revised version
> of the manuscript, we have addressed all of the comments in a way that we hope is to the satisfaction of the Editor and
> the Reviewers. In the revised version of the manuscript all the major changes, with respect to the
> original version of the paper, are presented in blue. The most significant changes to the original manuscript are as follows:
>
> 1) We have significantly revised Section 4. In particular, to make our results on the privacy-utility trade-off rigorous, we have kept the error term (that arises from the Taylor series of the probability of error) inside eq.(25) and eq.(26). Then, in Corollary 4.5, Corollary 4.9, and Corollary 4.13, we have computed the function in eq.(24) (or lower and upper bounds on it) for the Laplace mechanism, the Gaussian mechanism, and the generalized normal mechanism. These results now exactly and rigorously characterize the privacy-utility trade-off for the considered noise mechanisms.
>
> 2) For the generalized normal mechanism in Section 4.4, we now consider the $\epsilon$-DP (and not the $(\alpha,\epsilon)$-RDP as in the original version of the manuscript), which is defined in Definition 2.1. Such a metric is the strongest metric among those in the DP literature and hence, Corollary 4.13 now presents a stronger version of the privacy-utility trade-off than the one in Theorem 4.8 in the original version of the manuscript. We have also added Appendix L in which we derive the value of $p$ that minimizes the trade-off in eq.(35).
>
> 3) We have formalized and better explained a few parts that were deemed not clear by the Reviewers. For instance,  we have added the proof for the optimality of the maximum likelihood decoder on page 6, and we have used the $\ell_p, p>0$ sensitivity (and not the $\ell_1$ as in the original version of the manuscript) to describe the sensitivity property that the query function needs to satisfy.
>
> 4) We have added and better explained a few references. For instance, we have added the reference (Dwork et al. (2006b)) that introduced differential privacy, and we have added and discussed the reference (Liu (2018)) for the generalized normal mechanism in Section 4.4. We have also clarified that the result in eq.(27) in Proposition 4.4 and the result in Proposition 4.8 were already derived by Mironov (2017) and that we report their proof for completeness.
>
> 5) We have fixed a few points pointed out by the Reviewers. For instance, we have incorporated the parameter $\sigma$ whenever the randomized mechanism $\mathcal{K}$ is used (as defined in eq.(2)), and (on page 2) we have highlighted that the $(\alpha,\epsilon)$-RDP can be converted to the $(\epsilon,\delta)$-DP by using the result in (Mironov, 2017, Proposition 3).
> Finally, as suggested by the Editor,  we opted out of providing simulations on real-world data, as this paper is mainly of theoretical nature.

---

> > ### Comment · Action_Editors · 2022-06-19
> > **Thanks**
> >
> > Thank you to the authors for the prompt revision and attempt to address the authors' comments. We will now confer in private discussions and ask if there are any further questions.

---

### Decision · Action_Editors · 2022-07-10

**Recommendation:** Accept as is

**Comment:**

The reviewers were pleased by the significant revisions performed by the authors, which addressed their comments. While some reviewers still feel that the contributions are somewhat modest, they all nonetheless supported the paper for acceptance as-is.

Indeed, consulting the two guidelines for acceptance:
- Are the claims made in the submission supported by accurate, convincing and clear evidence?
- Would at least some individuals in TMLR's audience be interested in the findings of this paper?
the reviewers felt that both were true, and thus the paper should be accepted.